# At the Confluence of Artificial Intelligence and Edge Computing in IoT-Based Applications: A Review and New Perspectives

**DOI:** 10.3390/s23031639

**Published:** 2023-02-02

**Authors:** Amira Bourechak, Ouarda Zedadra, Mohamed Nadjib Kouahla, Antonio Guerrieri, Hamid Seridi, Giancarlo Fortino

**Affiliations:** 1LabSTIC Laboratory, Department of Computer Science, 8 Mai 1945 University. P.O. Box 401, Guelma 24000, Algeria; 2ICAR-CNR, Institute for High Performance Computing and Networking, National Research Council of Italy, Via P. Bucci 8/9C, 87036 Rende, CS, Italy; 3DIMES, University of Calabria, Via P. Bucci 41C, 87036 Rende, CS, Italy

**Keywords:** Internet of things, edge computing, artificial intelligence, machine learning, deep learning, swarm intelligence, big data analytics, bioinspired metaheuristic algorithms, resource management, edge analytics

## Abstract

Given its advantages in low latency, fast response, context-aware services, mobility, and privacy preservation, edge computing has emerged as the key support for intelligent applications and 5G/6G Internet of things (IoT) networks. This technology extends the cloud by providing intermediate services at the edge of the network and improving the quality of service for latency-sensitive applications. Many AI-based solutions with machine learning, deep learning, and swarm intelligence have exhibited the high potential to perform intelligent cognitive sensing, intelligent network management, big data analytics, and security enhancement for edge-based smart applications. Despite its many benefits, there are still concerns about the required capabilities of intelligent edge computing to deal with the computational complexity of machine learning techniques for big IoT data analytics. Resource constraints of edge computing, distributed computing, efficient orchestration, and synchronization of resources are all factors that require attention for quality of service improvement and cost-effective development of edge-based smart applications. In this context, this paper aims to explore the confluence of AI and edge in many application domains in order to leverage the potential of the existing research around these factors and identify new perspectives. The confluence of edge computing and AI improves the quality of user experience in emergency situations, such as in the Internet of vehicles, where critical inaccuracies or delays can lead to damage and accidents. These are the same factors that most studies have used to evaluate the success of an edge-based application. In this review, we first provide an in-depth analysis of the state of the art of AI in edge-based applications with a focus on eight application areas: smart agriculture, smart environment, smart grid, smart healthcare, smart industry, smart education, smart transportation, and security and privacy. Then, we present a qualitative comparison that emphasizes the main objective of the confluence, the roles and the use of artificial intelligence at the network edge, and the key enabling technologies for edge analytics. Then, open challenges, future research directions, and perspectives are identified and discussed. Finally, some conclusions are drawn.

## 1. Introduction

The adoption of new emerging technologies such as the Internet of things (IoT), wireless sensor networks (WSNs), cloud/edge computing, and 5G/6G communication networks in various fields (such as healthcare, agriculture, education, transportation, etc.), can bring many opportunities in improving people’s quality of life, thereby building intelligent systems that deliver high-quality, innovative services to the consumers. In the IoT environment, a large number of interconnected devices, such as sensors, mobiles, etc., lead to voluminous, heterogeneous, highly noisy, spatiotemporal-correlated and real-time data streams that need intelligent learning for efficient data analysis and meaningful insight extraction [1]. The success of any intelligent application can be attributed to the quality of the data collected, the effectiveness of data processing, storage, retrieval process, and the degree of accuracy and robustness of the data analysis results.

In conventional IoT solutions, the large amount of IoT data generated by the IoT devices is uploaded to the cloud via a wide area network (WAN) for further analysis to provide end-user feedback [2]. As the number of devices increases immensely, the communication costs, bandwidth, and latency become more expensive, which makes it unsuitable for real-time and time-sensitive applications. Furthermore, IoT has structured and unstructured, heterogeneous data that require advanced tools for its management. Fortunately, AI provides powerful tools for extracting valuable information to make accurate decisions in real-time [3]. Bringing AI closer to the edge offers a promising solution for achieving high system performance and improving quality of service (QoS) and quality of experience (QoE) for delay-sensitive applications.

Many AI-based solutions integrating machine learning (ML), deep learning (DL), and swarm intelligence (SI) have revealed a strong beneficial power to IoT applications in intelligent sensing [4], network management [5], resource management [6], big data analysis [7], and security system improvement [8]. New opportunities have emerged by adapting AI technologies to address the diverse characteristics of big IoT data features including volume, variety, velocity, veracity, and variability. DL models generate high-level abstraction and actionable insights that provide feedback through IoT systems to enhance their services [7]. The limitations of big data processing in the cloud and IoT systems, such as poor scalability, security issues, task allocation, fault tolerance, and low performance in conventional computing frameworks, can be resolved in a promising way by bioinspired computing [9].

Cloud-based infrastructures are considered the best suited to provide the needs of services and resources. Nevertheless, sending the massive data generated to the cloud poses some challenges, such as high latency, network congestion, and privacy issues. Fortunately, edge computing has emerged as a promising paradigm that enables computation at a location closer to the data source, which decreases the workload to the cloud, reduces latency, and improves privacy and the quality of service of smart applications and the user experience.

Edge computing is a distributed computing paradigm that extends cloud services to the edge of the network by deploying computational capabilities and storage between the terminal and the cloud devices. It addresses the limitations of cloud-based architecture by reducing bandwidth consumption, improving response time, and providing mobility and context-aware services. Fog computing, mist, cloudlet, and mobile edge computing (MEC), are all solutions belonging to the wider concept of edge computing [10].

The integration of edge computing along with artificial intelligence has the potential to gather, store, and process large amounts of IoT data, maximize the potential for rapid, real-time data analysis and decision-making, and deliver a variety of decentralized, low-latency, reliable, intelligent, and time-sensitive application services. Given the feature of resource-constrained and dynamic changes in edge computing, AI-based ML is considered the most suitable solution to maximize resource utilization, offload, and schedule computational tasks adaptively, dynamically, in real-time, and on-demand at the edge nodes, and meet application requirements in terms of time sensitivity and energy efficiency during computational task-sharing [11,12].

Several review papers have investigated the integration of AI in edge-based applications. In [13], Haddaji et al. present a comprehensive survey of AI techniques for security challenges in the Internet of vehicles (IoV). In this work, the authors evaluate the impact of AI on security in IoV. However, authors did not consider the enabling technologies and big data analytics. In [14], Laroui et al. cover the various use cases of IoT with edge and fog computing, job scheduling in edge computing, merging software-defined networking (SDN) and network function virtualization (NFV) with edge computing, security and privacy effort, and blockchain in edge computing. However, the authors did not consider the application of AI with the enabling technologies in smart applications. In addition, Chang et al. explore in [3] the combination of IoT and AI by using edge computing and the cloud. The authors focus on seven representative IoT application scenarios and specifically examine the techniques that enable the effective and efficient deployment of AI models. However, the authors did not consider the use of techniques of AI and the purpose and opportunities of applying AI in edge-based applications scenarios. On the other hand, Deng et al. concentrate in [15] on developing inference and training frameworks, by adapting models and hardware acceleration to support AI. Unfortunately, the authors did not consider the confluence of AI and edge computing in the different application domains of IoT. In [16], Xu et al. investigate the concept of edge intelligence from four axes: edge caching, edge training, edge inference, and edge offloading. In this review, the authors did not cover the confluence of AI and edge computing in the different application domains of IoT. In [17], the authors review ML techniques that are associated with three aspects of fog computing: management of resources, accuracy, and security. However, the authors did not consider the key enabling technologies for the deployment of AI models. In [18], the authors analyze the role of AI algorithms and the challenges of the application of these algorithms for resource management. However, the authors did not consider the key enabling technologies for the deployment of AI models. The authors in [19] provide a systematic review of nature-inspired approaches for resource management (task allocation, task scheduling, offloading) in cloud and edge computing. However, they do not address big data analytics.

In contrast to these reviews, our work investigates the confluence of AI and edge in many application domains by using different characteristics: AI, big data analytics, resource management, smart application, and enabling technologies. Table 1 shows a comparison of the existing surveys with our work. This table summarizes whether the corresponding surveys considered or not the characteristics used for the comparison.

Edge computing has become a promising solution for time-sensitive applications. However, the distributed, heterogeneous, and resource-constrained characteristics of edge computing pose many challenges and limitations in the design of on-device, distributed, and parallel computing in the edge infrastructure. This motivates us to write this review with a focus on exploring the proposed AI-based algorithms and their applicability in edge-based applications, investigating how AI can be used in edge-based IoT applications and how the confluence of edge and AI can improve QoS/QoE for many application domains, and highlighting the latest research and new technologies around this confluence.

In this paper we considered several points, which can be summarized as follows.

We reviewed 114 related papers that have been published from 2019 to present.To help readers understand the value and potential of implementing edge-based IoT infrastructure and to address cloud-based applications issues, we present an in-depth analysis of the state of the art of edge-based applications focusing on eight application areas: smart agriculture, smart environment, smart grid, smart healthcare, smart industry, smart education, smart transportation, and security and privacy.We present a qualitative comparison of related works in the eight aforementioned application areas. In this comparison we used eight characteristics: use case (the scope of application of AI for each application area), AI role (the potential of AI use), AI technique (AI-related algorithms), the used dataset, AI placement (on edge, cloud, or edge/cloud), employed technologies (technologies for running AI at the edge), the platform used for the implementation, and performance metrics. Three other columns are used to illustrate: the main contributions, benefits of edge-AI, and drawbacks of the reviewed works.We present a critical analysis of the presented state of the art by (1) exploring the current difficulties and limitations associated with the development and implementation of AI models and (2) investigating how AI can be used to overcome the difficulties presented by massive data in IoT systems and to improve the effectiveness of services on decentralized edge platforms.Based on the synthetic results, we suggest future trends for addressing the challenges of edge-based application deployment regarding big data analytics, scalability, resource management, security and privacy, and ultralow latency requirement.

The remainder of the paper is organized as follows. In Section 2, we review and qualitatively compare intelligent edge-based related works in eight application areas (i.e., smart agriculture, smart environment, smart grid, smart healthcare, smart industry, smart education, smart transportation, and security and privacy). Then, in Section 3, we present a discussion of the related works presented in Section 2. After that, we present in Section 4 current issues and future trends. We conclude the paper in Section 5. Figure 1 illustrates a schematic overview of the paper’s organization structure.

## 2. Artificial Intelligence in Edge-Based IoT Applications: Literature Review

Artificial intelligence techniques such as DL, ML, and bioinspired algorithms in IoT-based applications are necessary to manage the amount of data generated by various IoT devices, to process and analyze these data and, hence, to transform them into insights and be able to retrieve the knowledge required to make predictions, monitor, and make decisions.

In this section, we review the recent works on intelligent edge-based IoT applications. Furthermore, we present a qualitative comparison of the existing works in eight different application areas: smart agriculture, smart environment, smart grid, smart healthcare, smart industry, smart education, smart transportation, and security and privacy. The qualitative comparison is structured in tables by using several important characteristics: use case, main contributions, AI role, AI algorithm, dataset, AI placement, employed technologies, platforms and tools, performance metrics, benefits of the AI-Edge, and drawbacks. Table 2 shows the categorization of the reviewed works according to their application domain.

### 2.1. Smart Environment

Intelligent environmental monitoring aims to establish a full system that incorporates several types of sensors and IoT devices designed to measure various indications of the environment, such as temperature, humidity, and the concentration of pollutants in the air or the water. The integration of artificial intelligence and edge computing is essential to meet the requirements related to the complexity and the huge amount of environmental data that can be collected in this context. In the following, we first review and classify related works into four categories: air-quality monitoring (Section 2.1.1), water-quality monitoring (Section 2.1.2), smart water management (Section 2.1.3), and underwater monitoring (Section 2.1.4); then, we qualitatively compare these related works according to the aforementioned characteristics (see Table 3).

#### 2.1.1. Air Quality Monitoring (AQM)

For the optimal utilization of cloud resources and the improvement of computational power, a distributed fog computing framework for air-quality monitoring was developed in [23] by applying data preprocessing and clustering techniques to identify outliers on the fog layer by using the K-means algorithm and feeding only the relevant information to the cloud for the classification phase. This approach achieves *95%* accuracy with SVM compared to a multilayer perceptron (MLP), decision tree (DT), K-nearest neighbor (KNN), and naive Bayes (NB), and reduces the amount of data sent to the cloud still improving the response time.

In order to improve the computational efficiency and model performance of the environmental monitoring system considering regional characteristics when distributing various site monitoring models, the authors in [22] proposed a new framework called federated region-learning based on edge computing for PM2:5 air-quality monitoring. The authors first applied a regionalization algorithm that divides the monitoring locations into a set of subregions, each designed by microclouds in which the regional model is selected by the model that has the highest accuracy and, subsequently, the global model is aggregated by using two types of aggregation strategies to target the different bandwidth requirements better. The evaluation of the platform has been tried by using recurrent neural networks (RNNs) and convolutional neural networks (CNNs). It has been proven that the FRL approach improves the computational efficiency compared to the centralized training mode and normal federated learning (FL) [2].

In [24], Wardana et al. designed a distributed short-term air-quality prediction system for hourly PM2.5 concentrations based on a hybrid deep learning model composed of 1D CNN and long short-term memory networks (CNN-LSTM). They conceived an efficient posttraining quantization method to optimize the LSTM model and make it usable by resource-constrained edge devices wherein a one-dimensional CNN is used as a feature extractor. Through the results, the authors claim that the model has proven its performance in reducing execution time and latency.

In order to ensure privacy and reduce network traffic, the authors in [20] designed an efficient collaborative edge/cloud framework to predict the future concentration of fine particles in an individual space by selecting the best predictive model for the local edge based on its characteristics. The edge selects from the cloud the model with the highest correlation for a specific factor instead of choosing the model with the best performance. The performance of the system is validated with the LSTM algorithm for indoor PM10 and PM2.5 status prediction.

For efficient data generation and data privacy preservation for PM2.5 predictions, Putra et al. in [21] proposed a federated compressed learning based on an edge computing framework for massive-scale WSNs. This approach used compressed sensing techniques at the sensor level to reduce network data traffic. Then, at the fog layer, the data is trained distributively. After that, the global model is constituted by aggregating the local training models at the cloud layer. The evaluation is performed by using LSTM for PM2.5 concentration prediction and shows the efficiency of the compression sensing in reducing the data at the computation efficiency of the proposed model.

#### 2.1.2. Water Quality Monitoring (WQM)

In order to continuously monitor water quality in a distributed manner by using low-cost, cost-effective sensors, the authors of [25] developed an on-board sensor classifier for the detection of water pollutants. First, they used principal component analysis (PCA) algorithm to simplify and transform the original sensed data into a 3D space. Then, an adaptive classification scheme is employed on the transformed space to distinguish the contaminants by using a simple geometric model, the paramaters of which are learned by using a generational evolutionary algorithm (EA).

Authors in [26] developed a soft sensor model for real-time water-quality monitoring through intelligence at the edge to estimate the value of the biological oxygen demand. An edge/cloud platform is designed wherein the instance-based learning (IBK) algorithm is selected after a comparative study between different ML algorithms.

In [27], the authors proposed an online water-quality monitoring and early warning model based on edge computing. The authors proposed an improved backpropagation neural network (BPNN) by using a hybrid optimization method based on the Nelder–Mead simplex method and cuckoo search algorithm to optimize the weight and deviation of the BPNN.

#### 2.1.3. Smart Water Management (SWM)

In [28], the authors designed an efficient framework for water conservation based on blockchain technologies, soft computing, and machine learning. At the edge nodes (house nodes) a feed-forward neural network (FFNN) trained by symbiotic organism search is used to forecast the water consumption of each house. Then, the forecast value is compared to the historical value obtained by using a randomized probability distribution model for neural networks called the mixture density network (MDN). Based on these two calculated values, an incentive system is prepared in the blockchain to assign a good incentive to houses using less water than the historical value and applies a penalty to houses using more water than expected. Several factors were used such as (i) the number of people, (ii) the average income of the family, (iii) the profession of the members, and (iv) previous water demands. Results show the effectiveness of the approach for optimal water management.

#### 2.1.4. Underwater Monitoring (UWM)

Regarding marine environment monitoring, Yang et al. designed in [29] a fog/cloud-based framework for the effective management of ocean data and real-time monitoring of the marine environment. They introduced a fog layer to support data processing by using a numerical gradient-based method for data cleaning and an improved algorithm based on the evidence theory. This latter is used for multisensory information fusion with the aim of reducing the data volume and improving the data quality. In the cloud layer, a predictive model with BPNN is implemented. Authors argue that the framework can improve the efficiency of data use, improve the processing speed of ocean data and reduce the time delay. In [30], Lu et al. introduced a cognitive ocean network called motor anomaly detection system and detection of marine organisms. The proposed system consists of two methods: the first is deployed in the edge layer by using deep reinforcement learning and Raspberry Pi to prevent the default of underwater vehicles, and the second is deployed in the fog layer to detect marine organisms by using YOLO-based underwater method. Kwon et al. proposed in [32] a distributed DL approach based on federated learning with underwater IoT devices in the ocean environment. They used a multiagent deep deterministic policy gradient based on reinforcement learning (RL) to solve the problem of joint cell association and resource allocation in a way that improves the DL throughput of underwater IoT devices in underwater FL.

Regarding seawater quality prediction, Sun et al. developed in [31] a multivariate prediction model supported by edge computing for seawater quality assessment based on the combination of a PCA and relevance vector machine (RVM). Results show that the proposed model has higher prediction ability and less time consumption than other approaches.

### 2.2. Smart Grid

The integration of new technologies, such as IoT and artificial intelligence, into the power grid system allows (1) the design of a smart decision system support by developing an electricity distribution network. This offers the possibility of remotely measuring the state of the energy usage status online and thus enables the control of energy consumption and its further adjustment to the consumers’ energy needs. It also allows (2) the identification of abnormal behaviors in the consumption or production of electrical energy, and (3) the prediction of future electricity demand and energy consumption in an intelligent way based on the data acquired by the smart meters.

In this section, we present the recent works that use AI in edge-based smart grids and classify them into three categories: load/demand forecasting (Section 2.2.1), demand-side management (Section 2.2.2), and load-anomaly detection (Section 2.2.3). Moreover, we qualitatively compare the presented related works in Table 4.

#### 2.2.1. Load/Demand Forecasting (LDF)

Taïk and Cherkaoui proposed in [33] an edge-based, short-term individual load-forecasting framework. They used a distributed computation that uses an FL approach with the aim of addressing the challenges presented by the stochastic nature of consumption profiles and privacy in the smart grid. The realized simulations show that the approach outperforms the centralized model in terms of reducing the network load while preserving the privacy of the consumption data. This work does not solve the problem of detecting anomalies in the power consumption profile, which affects the accuracy of the model.

The authors in [34] proposed an edge-based short-term load-forecasting framework that uses an FL approach to enhance the prediction performance and reduce prediction errors. They proposed to group energy customers into similar users based on socioeconomic aspects or consumption similarities by using clustering techniques. This grouping of users is efficient, more effective than other trivial privacy-preserving schemes, and more adaptable to rapidly changing consumption patterns. In comparison with the centralized system, the proposed approach is more efficient in terms of model learning time, scalability, and inherently privacy-friendly alternatives. Furthermore, the communication overhead is reduced when energy-consumption measurements are recorded at a fine granularity.

Li et al. proposed in [35] a fog computing-based incremental learning for real-time day-ahead prediction of building energy demands. In order to choose the most suitable incremental machine learning model to address the high-speed real-time requirements of fog computing and generate good and fast edge intelligence, the authors compared two incremental learning algorithms, namely the swarm decision table (SDT) and the classical decision Hoeffding tree. Both combined with swarm feature selection to deal with the complexity of aggregated IoT and select only the significant features for efficient incremental machine learning. Results show the effectiveness of the proposed model.

Li et al. also proposed in [36] a fog computing-based platform for real-time prediction of electricity demand. First, a clustering algorithm is used to categorize users based on their total electricity consumption. Then, according to the characteristic of users’ historical electricity consumption, a predictive model using XGBoost or ARMA was selected. The accuracy of the proposed approach is *20%* higher in comparison to classical models.

In [37], Rabie et al. proposed a fog-based framework for accurate and fast electrical load forecasting in smart grids. First, a data summarization is performed on the collected data by applying several rules enabling the fog to send only the relevant data to the cloud by using fuzzy rank combined with a wrapper feature selection method and outlier detection. Then, an NB classifier is used to train the model and evaluate feature selection-based data processing techniques. Results show the effectiveness of the fog-based framework for accurate and fast load forecasting.

**Table 4 sensors-23-01639-t004:** Qualitative comparison of smart grid related works.

	Use Case	Ref	Contribution	AI Role (At the Edge)	AI Algorithm	Dataset	AI Placement	Employed Technology	Platform	Metrics	Benefits AI-Edge	Drawbacks
Smart grid	LDF	[33]	Short-term energy consumption forecasting	Prediction	LSTM	Pecan Street Inc’s Dataport site	Edge, cloud	Federated learning	Python, TensorFlow Federated 0.4.0 Tensorflow 1.13.1 backend	RMSE, MAPE	High accuracy	Heterogeneous data unsolved
		[34]	Short-term energy consumption forecasting	Prediction, classification	LSTM, K-means	Energy company UK Power Networks	Edge device, cloud	Federated learning	Python, TensorFlow	RMSE, training time	High accuracy, heterogeneous data solved	Privacy still low
		[35]	Day-ahead prediction of building energy demands	Prediction, Feature selection	Ant-bee, cuckoo, elephant, flower, genetic harmony, PSO, rhino, wolf, DT, HT	Ornl-research-house-3	Edge server (Raspberry Pi)	Low-cost model	Keras, Python	Accuracy, time, speed, MAE	High accuracy, low training time	Low interpretability
		[36]	Short-term electricity demand	Prediction, classification	XGBoost, K-means	Tianchi under license	Edge server (PC)	Low-cost model	Not mentioned	Training time, accuracy, cross-entropy loss	High accuracy	Data distribution unsolved
		[37]	Short-term electricity demand	Outlier detection, Feature selection, prediction	NB, wrapper FS, Filter FS	EUNITE dataset	Fog nodes		Matlab	Accuracy, error, precision, sensitivity/recall	High accuracy, reliability, resilience, stability	High complexity of model
		[38]	Online short-term energy prediction	data preprocessing, prediction	DNN	Real-world dataset	Edge server, edge devices, cloud	Collaborative learning	Not mentioned	Flexibility, accuracy	Flexibility, high accuracy, dynamic data, IoT addressed, real-time prediction	Less scalability
		[39]	Load forecasting for optimal energy management	Prediction	CNN	IHEPC dataset	Edge devices	/	TensorFlow, Keras	MAPE, RMSE	Low complexity	Heterogeneous data, uncertainties, privacy is not addressed
		[40]	Online short-term residential load forecasting	Prediction	STN	Ohta-AMPds datasets	Edge device	Low-cost model-reservoir computing	Not mentioned	RMSE, MAE	Low complexity, high accuracy	Heterogeneity not addressed
	D.S.M	[41]	Demand-side management	Resource management	RL	Real-world dataset	Edge server (Raspberry Pi)	–	Real implementation	Not mentioned	/	Less scalability
		[42]	Demand-side management	Classification	LDA	REFIT project	Edge server	Low-cost model	Not mentioned	MAPE, RMSE		
		[43]	Managing prosumers over wireless networks	Data preprocessing, prediction	LSTM	Pecan Street Inc.’s Dataport site	Edge server	Federated learning	TensorFlow	RMSE, data transmitted	Heterogeneous data addressed, high accuracy low-communication cost	Single-point failure not addressed
	LAD	[44]	Detection of anomalous power consumption at household	prediction	GBR, RFR, LR, SVR	IHEPC dataset	Edge server, fog	/	Not mentioned	MAPE, RMSE	Load reduction	Communication cost still high
		[45]	Anomaly detection in smart-meter data	resource allocation, classification	SDA, GA, kNN	IHEPC dataset	Edge server	/	Not mentioned	Accuracy, execution time, energy consumption	–	–
		[46]	Electric energy fraud detection	Dimensionality reduction, prediction	DTR, LR	D1C database	Edge server Raspberry Pi model	–	Not mentioned	MAPE	–	–
		[47]	Anomaly detection consumption smart grid	Classification	DNN, HDBSC K-means, KNN	Midwest region	Edge server, Raspberry Pi	/	Not mentioned	Testing time, frequency, model size	Low complexity, high accuracy	–
		[48]	Energy theft detection	Feature-extraction classification	VAE-GAN, K-means	GEF Com 2012 public dataset	Edge server	/	Not mentioned	ROC curve, running efficiency	Adaptive model, high accuracy	-
		[49]	Energy theft detection	Classification	(SGCC) dataset	Edge devices	Federated learning	Flower	RMSE, log loss accuracy, precision F-measure	Privacy		Low accuracy compared with the centralized model

Luo et al. proposed in [38] a short-term energy prediction-based edge computing platform. It consists of four stages: (1) data acquisition and fusion performed on edge nodes to support redundant multisource heterogeneous IoT by using a semantic information model, (2) event data generating stage performed in the routing nodes to deal with the weak semantics of IoT data, (3) local aggregation performed on edge nodes in order to aggregate data based on its spatiotemporal semantics, and (4) a prediction model built in the central server by using an online deep neural network model which updates the prediction model in real time over the stream of data instances to accommodate the changes in the IoT environment.

The uthors of [39] proposed a short-term energy consumption forecasting model named Energy-Net, optimized for the deployment on resources constrained devices. Energy-Net uses a deep learning approach that exploits the spatial and temporal learning capability for the prediction of energy consumption.

In [40], the authors proposed a framework based on edge computing for short-term residential electricity demand forecasting by using online learning and reservoir computing by state network architecture to avoid high computational costs considering the nonlinear and dynamic behavior of demand time series improve the accuracy of the prediction model by continuously tracking the dynamically changing demand characteristics.

#### 2.2.2. Demand-Side Management (DSM)

Cicirelli et al. proposed in [41] an edge-based energy management system to reduce the energy cost of daily household appliances. They proposed a load appliance scheduling algorithm that exploits reinforcement learning. It takes into account time variable profiles regarding energy cost, production of energy, and energy consumption of the appliances. The approach is validated through the implementation of a real-world use case that shows convincing results.

Tom et al. used in [42] a fog-based IoT architecture to design a smart energy management system and build a solution for demand reduction of individual houses in a locality during peak hours. They used autoregressive integrated moving average (ARIMA) to predict consumer utilization by studying consumers’ daily usage patterns and a discriminant analysis to find the appliances playing a significant role.

Taik et al. proposed in [43] a multilevel prodecision framework based on federated learning for intelligent decision-making in energy markets. It prioritizes individual prosumer decisions supported by the 5G wireless network for rapid coordination between community members. Each prosumer forecasts energy production and consumption to make proactive business decisions taking into account collective-level demands. The result achieves high accuracy for different energy resources with low communication costs.

#### 2.2.3. Load Anomaly Detection (LAD)

For providing real-time anomaly detection for solving big data issues in the power consumption domain, Jaiswal et al. [44] proposed a hierarchically distributed fog computing architecture for smart meter data analysis in households by using an ensemble method consisting of four lightweight regression models: linear regression (LR), support vector regression (SVR), random forest regression (RFR), gradient boosting regression (GBR).

Liu et al. designed a distributed fog computing platform for detecting smart meter data anomalies [45]. They used a stacked denoising autoencoder and KNN classifier deployed on the fog nodes. At the same time, an adaptive elitist GA is used to optimize the required computational task for supporting the model in the fog nodes and minimizing the communication cost.

Olivares–Rojas et al. proposed a detection of electric energy fraud supported by edge computing [46]. First, a dimensionality reduction by the PCA algorithm is used. Then, prediction techniques based on previously established patterns of energy consumption/production by LR, DT, neural networks, and MLP are performed.

Utomo and Hsiung developed in [47] a multitiered solution for efficient and fast real-time anomaly detection. They use a clustering model based on the combination of the K-means and hierarchical density-based spatial clustering of applications with noise (HDBSCAN) for data reduction. Then, the oversampling mechanism SMOTE is used to cover the imbalanced dataset. The authors compare support vector regression (SVR), KNN, and DNN to choose the best detector anomalies classifier.

In [48], Zhang et al. proposed a framework supported by the edge for energy theft detection. The detection passes through three stages: (1) feature learning based on load profile for energy consumption analysis is implemented by using VAE-GAN, (2) k-means clustering is used to determine the representative features of normal load profiles, and (3) abnormality degree is calculated by using a threshold-based abnormality detector.

In [49], the authors proposed a federated voting classifier for energy theft detection. The authors used a majority voting for the three classifiers (i.e., RF, KNN, and bagging classifier (BG)). Results show the effectiveness of the model compared to the centralized cloud model in terms of privacy.

### 2.3. Smart Agriculture

The integration of IoT technologies and edge computing creates great opportunities for the agricultural field. It makes up a support system that is able to monitor, capture and analyze information about crops and livestock in real time. It may include early plant disease prevention, better soil monitoring and management, livestock management, and reduction of environmental impacts by climate change prediction. The use of artificial intelligence improves the production process, maintains the highest levels of crop quality, and reduces costs and waste.

In this Section, we review and classify related works into five categories: weather prediction (Section 2.3.1), livestock management (Section 2.3.2), smart irrigation (Section 2.3.3), crop monitoring and disease detection (Section 2.3.4), and monitoring the health status of agriculture machines (Section 2.3.5); then, we qualitatively compare them in Table 5 according to the aforementioned characteristics.

#### 2.3.1. Weather Prediction (WP)

Guillén et al. consider in their work [50] the construction of an automated decision-making framework for precision agriculture. In such problems, constraints including low-bandwidth connectivity and energy consumption must be addressed. To this end, the authors proposed an edge-based platform for the early identification of frost on crops by estimating the low temperatures through an LSTM model on edge devices. This helps farmers to obtain a temperature prediction in real time. The proposed model is evaluated in terms of performance and power consumption of edge devices.

In [51], Kaur and Sood proposed a framework for drought forecasting. At the fog layer, a dimensionality reduction method based on PCA is used, although the classification of drought severity is performed on the cloud layer by using ANN with genetic algorithms (GA). After a fixed interval of time, the predicted values of drought severity are used by the ARIMA model for future drought forecasting.

#### 2.3.2. Livestock Management (LM)

The authors of [52] suggested strategies for offloading computation from cloud to fog to assist the huge quantity of multimedia data from IoT devices in smart agriculture. They process more deep learning tasks at the fog layer by assigning the maximum number of layers on each fog node with the aim to (1) reduce the amount of data transferred to the cloud, (2) utilize resources efficiently, and (3) reduce network congestion. The authors show, through experiments, that the proposed strategies had satisfactory results in terms of bandwidth, number of deep learning tasks for each node, and the data volume transferred to the cloud compared with existing methods.

For accurate and early detection of lameness in smart dairy farming, Taneja et al. developed in [53] an application based on fog/cloud computing to collect activity data, monitor the cattle in real time, and identify lame cattle at an early stage. They employed a K-means algorithm at the fog layer for data processing, and classification was done on the cloud by using the KNN algorithm. Results show that the application can detect lameness three days before it can be visually captured by the farmer with high accuracy and minimal communication cost.

#### 2.3.3. Smart Irrigation (SI)

To improve irrigation water, Cordeiro et al. have proposed in [54] a fog-based framework for soil moisture forecasting. First, a KNN data imputation is used for the missing values to increase data reliability. Subsequently, an LSTM is used for the prediction by employing a small single-board computer.

In [55], authors proposed a low-cost intelligent irrigation system based on edge computing to forecast environmental factors. They used an LSTM/gated recurrent units (GRU)-based model for a comparative analysis by using many frameworks. Results show the reliability of LSTM and GRU for the prediction of environmental factors.

#### 2.3.4. Crop Monitoring and Disease Detection (CMDD)

Identifying crop diseases is one of the most difficult tasks in smart agriculture. We present below some recent and relevant related works.

A timely detection on crops to stop diseases from spreading was presented in [56]. The authors proposed a model named deep leaf, a coffee plant disease detector based on edge computing. It detects the main biotic stresses affecting crops. The proposed model uses a dynamic compression algorithm based on K-means for the reduction of a model footprint to reduce the complexity of the CNN model and run it on devices with limited hardware capabilities.

Likewise, the authors of [57] proposed an IoT monitoring framework for detecting tomato diseases. First, a pretraining model is constructed on the cloud by using VGG networks. Then, in order to fit the model on embedded mobile platforms, a depth-wise separable convolutional network is used to reduce the parameters of the model and calculation of model feature extractor. The experimental results show that the framework can accurately detect crop diseases in less time.

Zhang and Li proposed in [58] an adaptive sensing strategy for the crop life cycle based on edge computing. First, the growth stage of the crop is divided by the Gath–Geva fuzzy clustering for the sensing nodes. Then, data-driven algorithms are used in the edge server to extract and optimize the key parameters corresponding to the growth stage in order to increase the data values by reducing redundancy and improving the correlations between the sensing data. Finally, a neural network-based crop growth stage prediction model is performed.

#### 2.3.5. Monitoring the Health Status of Agriculture Machines (MHSAM)

Gupta et al. proposed in [59] an edge-based framework for agriculture vehicle health monitoring by using ANN. To decrease the model’s complexity in terms of computing and develop a lightweight one that can be deployed on a smartphone, two levels of optimization using a genetic algorithm for ANN are conducted.

In [60], Rajakumar et al. proposed a framework to identify the health condition of the vehicles. They design a fault-detection algorithm by using a deep convolutional neural network (DCNN) on smartphones. The authors used the Levy flight optimization algorithm (LFOA) to optimize the network structure of the DCNN, minimize the number of neurons in the DCNN hidden layer, minimize the number of input features from the audio recordings, and enhance the classification accuracy.

### 2.4. Smart Education

Smart education is defined as the integration of IoT devices with learning that can establish location information, motion sensing, and visual recognition tools. IoT devices in combination with other technologies such as artificial intelligence and cloud computing are used to evaluate educators’ engagement and skills and improve the teaching and learning expertise in the field. Using edge computing in smart education: (1) reduces the delay, (2) improves the level of service delivery for learners by protecting information transmitted, (3) and guarantees that every communication process is managed effectively [134]. In this section, we review and classify related works into two categories: student engagement monitoring (Section 2.4.1) and skill assessment ( Section 2.4.2); then, we present a qualitative comparison of related works in Table 6.

#### 2.4.1. Student Engagement Monitoring (SEM)

Umarale et al. proposed in [61] an edge computing-based deep learning technique for detecting and identifying the attention level of learners within online learning sessions. They employed, on edge devices, a lightweight CNN model that uses facial image data to determine the attention level. The output is further processed on the cloud to derive an attention average of the participants. Then, the attention average is reported to the host, helping the teachers to obtain information about the students’ performances and further helping them identify the students who were inattentive during the session.

Li et al. designed in [63] a real-time intervention system for negative emotional contagion in the classroom based on edge computing infrastructure. The system integrates an emotional contagion model with a deep learning algorithm. To achieve multiperson emotional recognition, an embedded device is used to process images to recognize the emotions of all the students in the classroom and locate the source of the negative emotion to take real-time intervention actions through visual emotion identification.

In [64] Preuveneers et al. introduced a learning management system for engagement monitoring by using a collaborative edge-cloud framework. They combine FL with secure multiparty computation to process users’ behavior data to analyze student involvement and increase the online learning system to the next level.

In [62] to enhance students’ independence in resolving difficult engineering problems and boost their marketability, authors created an experimental open-source distance learning platform based on edge computing and artificial intelligence that is well-suited for distance learning.

The authors in [65] proposed a framework for monitoring student stress and generating real-time alerts to predict student stress. The authors used Visual Geometry Group (VGG16) for facial expression, bi-LSTM for speech texture analysis, and multinomial NB techniques to generate emotion scores and classify stress events as normal or abnormal.

#### 2.4.2. Skill Assessment (SA)

Sood and Singh proposed in [66] an e-learning framework with multiple functional aspects. The proposed framework helps in enhancing the skill set of students. The first aspect is that of monitoring the academic skill data of learners in order to classify their employability at the early stage of graduation. The second aspect consists of skill-set assessment based on clustering to improve their required skill set through e-learning. Finally, an adaptive resource usage elasticity prediction is made. Experimental results show that the proposed approach achieves 96.45% accuracy of classification.

By utilizing the information gathered by IoT devices to make smart decisions about the quality of education and the academic environment, Ahanger et al. [67] developed an intelligent framework based on hybrid cloud/fog infrastructure for education quality assessment. They proposed a model based on the Adaptive Neuro-Fuzzy Inference System (ANFIS) for decision modeling based on the education quality scale determined by classification at the fog layer. Results show the effectiveness and reliability of the model with good accuracy about the quality of education access compared to the most recent decision models.

The authors of [68] presented a particle swarm optimization (PSO)-driven edge computing method that might aid in the cooperation and optimization of various ideological and political course resources in a mobile edge computing 5G network for intelligent education assessment on ideology and politics. The authors define the optimization issue as reducing the worst-case energy consumption in task offloading, as well as the decision-making and resource allocation of task offloading supported by edge caching. The outcomes of the experiment show that the suggested method achieves a high level of experience and energy conservation.

### 2.5. Smart Industry

Edge computing and AI play an effective part in the automation of large-scale industrial processes by providing efficient distribution of applications and an intelligent deployment strategy that provides ideal service delivery to users and customers. For the intelligent industry, fast and real-time detection of machine malfunction and preservation of product quality are very important. In addition to providing high-quality commercial operations in the industrial sector, the management of product provides suitable actions to prevent wastage of the products and high service delivery, whereas in the field of the finance industry, by constructing a smart financial technology application, banks and financial institutions may provide quality services to their consumers via individualized virtual supervision [69]. We first review and classify related works into four categories: financial industry (Section 2.5.1), commercial industries (Section 2.5.2), machine malfunction monitoring (Section 2.5.3), and product quality monitoring and prediction (Section 2.5.4); then, we present a qualitative comparison of related works in Table 7.

#### 2.5.1. Financial Industry (FI)

Manusami et al. developed in [69] designed a ranking-based strategy to classify financial tasks arriving at the edge according to their priority as risky and nonrisky tasks. So as to minimize network energy consumption, the ranked financial tasks are assigned to appropriate computing devices for further analysis by using a service-deployment mechanism based on a perfect matching theorem in graph theory. Subsequently, SVM is used to analyze the ranked tasks at the edge networks for immediate prediction and detection of fraud.

An early warning model for financial risk prediction of the enterprise based on MEC is proposed in [70]. The authors used an optimized BPNN with an edge service preloading optimization model which is applied based on the information obtained about the geographical information related to points of interest and BPNN. Then, according to the user’s location feature vector, the probability of the user’s next service is predicted. Results show that the service preloading optimization based on the geographic information points and BPNN improves the response speed.

#### 2.5.2. Commercial Industries (CI)

In [71], Neelakantam et al. designed a fog computing framework for product demand forecasting and decision-making. They used PCA and K-means for clustering products based on product demand and grouped products into three categories, namely, low, medium, and high demand. Then, the reinforcement learning model is used for product distribution decision-making.

#### 2.5.3. Machine Malfunction Monitoring (MMM)

The authors in [72] proposed a framework based on fog computing to analyze and classify the machine sounds in order to monitor and identify the malfunctioning machines. To extract the important features of the audio signal, the authors used linear prediction coefficients (LPC) and melfrequency cepstral coefficients (MFCC). Then, they used supervised machine learning models (such as RF, SVM, AdaBoost Classifier, and MLP) to detect and classify the malfunctioning machine sounds as normal and abnormal. These models showed their performance in detecting low-level sound from the audio signal and enhancing the service time.

Syafrudin et al. proposed in [73] an edge-based fault detection by using density-based spatial clustering for outlier detection and for covering the imbalanced data issue. The oversampling SMOTE method is used, whereby an RF algorithm is applied in prediction. The proposed method achieves higher accuracy and fast fault detection.

In [74], Fawwaz and Chung proposed an edge-cloud framework for real-time fault detection based on combined LSTM-AE algorithms. This handles both multivariate time series and noisy data. First, a novel correlation and redundancy-aware feature selection (CRFS) approach by a genetic algorithm is implemented. Then, a pretrained model is conducted on the cloud with the combination of LTSM and AE. Secondly, the pretrained model is transferred to the edge for real-time fault detection. Experimental results show the effectiveness of the model by achieving shorter detection times, better accuracy, and more robust performance in the presence of noisy data.

Park et al. [75] developed a model for real-time machine fault detection in smart manufacturing. A lightweight LSTM is developed for an edge device and a Raspberry Pi for implementation. Results show that the model outperforms the existing models.

Li et al. [76] designed collaborative fog-cloud computing for inspection manufacturing by using CNN with offloading strategies. These latter offload the low layer of CNN to the fog nodes. For fast detecting defects in a product and identifying its degree, an early exit strategy is used. The proposed method reduces the data transmitted to the cloud and hence can perform real-time detection.

#### 2.5.4. Product quality monitoring and prediction (PQMP)

Feng et al. [77] proposed an edge-based assembly quality prediction in an industrial IoT environment. They used an RF for feature selection while the SMOTE–Adaboost method with jointly optimized hyperparameters was used for imbalanced classification. The experimental findings demonstrate that, in terms of predicting assembly quality, the suggested technique is more accurate than existing classification methods.

In [78], the authors proposed a fog-based framework for tool wear monitoring and prediction. First, the authors used both CNN and LSTM to extract tool wear temporal features on fog nodes. Then, a bidirectional LSTM model (BiLSTM) is performed on the cloud for tool wear prediction based on the features extracted by the MCLSTM model. Results show the effectiveness of the model in terms of high monitoring accuracy and low response latency.

For real-time and efficient processing tasks in smart production lines, Wang and Li [79] proposed a hybrid heuristic algorithm, an improved particle swarm optimization (IPSO) algorithm, and the improved ant colony optimization (IACO) for task scheduling in fog computing in order to solve the problem of end devices with low computational power and significant energy use.

### 2.6. Smart Healthcare

IoT, AI, and edge computing paradigms are considered major keys to the new revolution in healthcare by providing an intelligent system that aims at improving the quality of care services such as (i) remote physical patient monitoring, and (ii) automatic diagnosis and detection of diseases at early stages. In this section, we present the existing recent works in intelligent edge-based healthcare applications. In particular, we review and classify related works into five different categories: diet health management (Section 2.6.1), ambient assisted living (Section 2.6.2), human activity recognition (Section 2.6.3), location-based disease prediction (Section 2.6.4), and disease diagnosis (Section 2.6.5; then, we qualitatively compare them in Table 8.

#### 2.6.1. Diet Health Management (DHM)

One of the main reasons for health damage is an unhealthy diet. To tackle the automation of dietary assessment, authors in [80] proposed a food-recognition model with a deep residual convolutional neural network, which determines whether the food photos include enough vegetables. In order to make predictions on a mobile device without connecting to a cloud server, the authors quantized the network weights of the proposed model by using posttraining quantization methods into low-bit fixed-point representations.

Likewise, Liu et al. [81] proposed a DL-based food recognition for assessing diets. Taking into account the limited computation resources and low battery life on mobile devices, the preprocessing and segmentation of food images have been performed on edge devices (smartphones). At the same time, the classification with a pretrained GoogLeNet model for feature extraction and softmax classifier was done on a cloud server. The model exceeds other works in terms of accuracy, with a quicker response time and reduced energy use, according to experimental results.

#### 2.6.2. Ambient Assisted Living (AAL)

For accurate and timely fall detection, the authors of [82] developed an intelligent system based on fog/cloud computing architecture. The cloud data analysis resources are used to train the hybrid DL model (GRU/LSTM), whereas the DL model inference is implemented on a fog smart gateway for real-time fall detection and alert notification to caregivers’ smartphones. To overcome the complex challenges of resource limitations on the fog for DL inference, an efficient and automatic deployment is performed by using virtualization technologies. Results show how well the system works for providing quick, precise responses and enhancing customer service.

For elderly patients with chronic disease monitoring, Hassan et al. proposed in [83] a fog/cloud framework. A firefly algorithm (FA) was used to optimize the NB classifier by selecting the minimal features that yield the highest accuracy. The framework collected data from the elderly patient by using ambient and biological sensors, fused the data into contextual states, and utilized context-aware algorithms to forecast the patient’s health status in real time. The introduced framework includes a five-phase classification method to handle huge datasets that are unbalanced as a result of elderly patients being followed for an extended period of time.

In [84], authors proposed a framework for real-time fall incident monitoring by using ML algorithms based on fog computing. First, they used linear discriminant analysis (LDA) to reduce the dimensionality of extracted features. Then, they employed SVM and KNN for classification.

Divya et al. [85] proposed a fall detection framework. It consists of four layers: edge devices, mist, fog, and cloud. The edge consists of a smart camera, which deploys a compressed DNN model for fall detection. Basic data filtering and rule-based decision-making are handled by the mist. Images are transmitted to the cloud storage only when a fall is detected, and the edge detection output is only delivered to the higher fog layer if a fall is observed. Xtreme gradient boosting and RF methods are used to build the model in the cloud.

The authors of [86] designed a cloud/edge-based federated learning framework for in-home health monitoring named FedHome. The authors used a lightweight convolutional generative autoencoder to deal with the unbalanced and non-ID distribution health monitoring data with high accuracy in predictions.

#### 2.6.3. Human Activity Recognition (HAR)

The authors of [87] introduced a light DL framework that uses SMOTE to solve the problem of imbalance labels and implemented a CNN embedding feature (CNNEF) to understand abnormal human activities through the sensor data in edge nodes to predict the user’s behavior, detect anomalous activities, and offer more accurate, efficient, and real-time services. Then, the extracted high-level embedding features from CNNEF are given to the classical ML algorithms, such as logistic regression, KNN, DT, NB, RF, and SVM.

A brand-new DL-based human activity recognition framework for edge computing termed DL-HAR was suggested in [88]. The proposed framework seeks to accelerate decision-making. It employs a DL algorithm to cut down on communication with the cloud servers, cutting down on potential delays and round trips. In order to detect the activity time-series data coming from sensors or smartphone devices, the framework first trains the DRNN model on the server side because of its high capacity and then transmits the image of the learned DRNN model to Docker containers on Raspberry Pi3 edge devices.

In [89], the authors proposed an edge-based framework for human activity recognition designed for wearable edge devices. The authors design an energy-efficient solution by using an adaptive CNN that selects a portion of the baseline architecture to use during the inference phase instead of using the full architecture.

The authors of [90] proposed a blockchain based on a fog monitoring system to identify human activities as an interface of e-healthcare services. The proposed framework categorizes and classifies the video frames based on patient activities by using the SVM algorithm. Videos of various human activities are retrieved by using a multiclass cooperative categorization approach to increase the activity classification accuracy in video features, which are then processed into action vocabulary for efficiency and accuracy. In a similar manner, an SVM based on the error-correction output codes (ECOC) architecture is used to classify activities.

A Bayesian deep learning network, which aids in inferring and accurately identifying various physical data acquired from individuals to track their physical activities, was examined by the authors of [91] by utilizing edge computing. The effectiveness of this wearable Internet of things system with multimedia technology is then assessed by using the results of some experiments and analyzed in terms of accuracy, efficiency, mean residual error, delay, and energy consumption.

In order to anticipate health conditions in real-time based on an individual’s physical postures, the authors of the paper in [92] developed a fog/cloud system. In this study, they use the continuous time series policy to store anticipated activity ratings on the cloud and give future health references to accredited medical professionals. The physical abnormality that is predicted and the level of health severity are closely correlated with the issuance of the warning. Clear benefits of fog analytics over cloud-based monitoring systems include an improvement in the recognition rate of up to 46.45% for 40 FPS and 45.72% for 30 FPS. By attaining high activity prediction accuracy and low latency, the computed results demonstrate why the proposed fog analytics monitoring system is preferable to other cloud-based monitoring solutions.

#### 2.6.4. Location-Based Disease Prediction (LDP)

Ahanger et al. developed in [93] a fog/cloud framework to forecast COVID-19 cases, employ user-held devices, and track the disease’s spread. First, to identify contaminated individuals and areas, the authors used fuzzy C-mean classification. Then, in order to predict the possibility of COVID-19 symptoms in the geographical patterns, the authors used a temporal recurrent neural network. The self-organization mapping (SOM) method is used to present data on geolocations for COVID-19 dynamical behavior over spatial–temporal domains.

The authors of [94] proposed a fog-cloud framework for remote diagnosis of ENCPH spread based on the patient’s health symptoms and the surrounding environment. The fog layer analyzes a patient’s category based on parameters from health-related data by using a fuzzy C-Means classifier. At the same time, the prediction model based on spatiotemporal domains that use T-RNN is used to manage the medical resources. A SOM technique is used for outbreak geographic visualization.

A novel fog computing-based e-Healthcare framework was presented by Majumdar et al. in [95] to monitor KFD-infected patients throughout the early stages of infection and manage the disease epidemic. A new extremal optimization tailored neural network classification technique has been created by employing the hybridization of the extremal optimization with the feed-forward neural network in order to guarantee a high prediction rate. A location-based alert system has also been recommended to give each KFD-infected user’s location information based on their GPS location as well as the locations of risky areas as soon as possible in order to prevent the epidemic.

A fog-assisted cloud-supported healthcare system was created by Vijayakumar et al. in [96] for the real-time identification and prevention of illnesses spread by mosquitoes. The categorization of illnesses spread by mosquitoes has been done based on symptoms. The registered user is divided into infected and uninfected groups by using a fuzzy KNN algorithm. Social network data is examined to identify risky regions. Alert messages have been sent to registered users in an attempt to avoid an epidemic so they may stay away from risky locations.

The authors of [97] designed an edge-cloud collaborative learning framework for the local diagnosis of COVID-19 by using the VGG16 algorithm. The authors used a clustering federated learning approach in order to solve the heterogeneity and the divergence in the data distribution.

Singh et al. developed in [98] a fog-based QoS framework to monitor the state of health of citizens and prevent and ensure safety from COVID-19. The fog layer provides real-time processing of users’ health data in order to predict COVID-19 infection. The unique patient identification, which is made up of patient data and geographical information, is then transferred to the cloud layer for further processing when the diagnosis is positive. The results of the experiments show that the proposed model is very efficient for remote diagnosis of COVID-19 infection and may be utilized as a time-saving substitute for labor-intensive clinical diagnostic procedures.

Singh et al. developed in [99] a collaborative edge/cloud framework for remotely diagnosing COVID-19. For the purpose of easy deployment on low-powered mobile devices and devices and quick diagnosis, they used an optimized DL model inspired by the MobileNet V2 model architecture. The model was first trained on the cloud; then its backup was sent to edge devices to perform the diagnosis of COVID-19 infection. Finally, when the diagnosis is positive, the unique patient identifier composed of patient information and location information is sent to the cloud layer for further action. Experimental results demonstrate that the proposed model is very effective for remote diagnosis of COVID-19 infection and can be used as an efficient alternative to time-consuming clinical diagnostic tests.

In [100], the authors proposed an intelligent health monitoring framework, iCovidCare for the prediction of coronavirus disease based on an ensemble RF model. First, a rule-based approach is employed at the local device to diagnose the coronavirus disease based on the temperature sensor data. Then at the cloud server, the feature selection, and fusion are applied for COVID-19 disease prediction.

#### 2.6.5. Disease Diagnosis (DD)

In order to achieve an early and accurate diagnosis and detection of lung cancer while maintaining privacy, low latency, and mobility, Prabukumar et al. developed in [101] a fog-based system for the diagnosis of lung nodules. First, fuzzy hybrid C-Means and region-growth segmentation algorithms were used for image segmentation and feature extraction. Then, cuckoo search and SVM were used for feature selection and classification, respectively.

A paradigm for intelligent patient monitoring of cardiomyopathy patients by using sensors and wearable technology is presented by the authors in [102]. By relocating sensors in the monitored region, a fuzzy Harris hawks optimizer (FHHO) is first utilized to expand the coverage of monitored patients, and then a wearable sensing data optimization (WSDO) algorithm is employed for heart rate detection. The experimental findings show that the optimized model is successful in terms of the number of sensors used, accuracy, and response time, as well as sufficient patient coverage.

A real-time smart remote monitoring system for patients with chronic illnesses was suggested by the authors in [103]. Four layers make up the suggested framework: the sensing layer for data collection, the edge device layer for offline preprocessing, the edge server layer, and the cloud layer for further online operations. For the purpose of forecasting the patient’s health status in dispersed emergency occurrences, the offline classification techniques are trained in the cloud. The whale optimization algorithm (WOA) and NB are used in the suggested technique to choose a small collection of features with a high level of accuracy.

The authors of [104] proposed an ensemble approach based on data fusion in fog computing by using medical data from body sensor networks (BSNs) for heart disease prediction. For their classification technique, they included a number of temporal and frequency domain characteristics into a kernel RF ensemble. To create higher quality data that is input to the ensembles for heart disease prediction, data from many sensors is fused.

The authors of [105] proposed an adaptive neuro-fuzzy inference system model for Parkinson’s disease prediction. The fog takes a prominent role in feature extraction from IoT sensors and provides the principal functions. Then, the parameters of the model are adjusted through grey wolf optimization (GWO) and PSO. Results show that the proposed model succeeds in predicting Parkinson’s disease with good accuracy.

Shynu et al. developed in [106] a fog computing-based framework for disease prediction. First, for the protection and effective data storage and data sharing, a blockchain in the fog nodes is used. The patient data for patients with diabetes and cardiovascular disease are then initially grouped by using a rule-based clustering method. Finally, a feature selection-based adaptive neuro-fuzzy inference system is used to predict diabetes and cardiovascular illnesses (FS-ANFIS).

In order to provide low-latency responses in identifying emergency situations for cardiac patients, Cheikhrouhou et al. proposed in [107] a remote cardiac patient monitoring based on hybrid fog-cloud architecture for analyzing ECG signals captured from IoT wearable devices. Results show that the proposed approach based on a one-dimensional CNN approach for arrhythmia cardiovascular disease detection could achieve an accuracy of 99% with 25% improvement in the overall response time.

Similarly, for real-time physiological data analysis, the authors in [109] designed a framework for health monitoring based on fog computing. The system consists of three layers. The first is the wearable layer wherein an RK-PCA is used to eliminate erroneous data. A fog layer, which consists of an onlooker node is used to eliminate redundant data generated by wearable devices and health status prediction. Then fog nodes for health status detection. Finally, there is a cloud layer for data storage. In addition, a multiobjective optimization algorithm is used to solve fog overloading in smart healthcare applications. Experimental results show the stability of the system compared to the cloud-based approach, while less latency, execution time, a high detection accuracy are improved.

In [108], the authors proposed a deep learning model to be supported by edge computing and investigated it in the diagnosis for identification of heart disease from the data collected by using IoMT devices. The proposed effective training scheme for DNN (ETS-DNN) model incorporates a modified hybrid water wave optimization technique to tune the parameters of the DNN structure.

To improve the detection of impending hypoglycemia, the authors of [110] developed an embedded deep-edge learning model by using evidential regression and attention-based recurrent neural network for real-time blood glucose.

### 2.7. Smart Transportation:

The use of IoT and AI technologies in the transportation field consists of collecting information about vehicles, drivers, and roads with the objective of creating a real-time traffic management system by performing traffic road condition monitoring, detecting events in real time for traffic safety, and preventing perturbations that impact on traffic flow and parking availability.

In this Section, we review and classify related works into three categories: smart parking management (Section 2.7.1), traffic monitoring/prediction (Section 2.7.2), and intelligent transportation management (Section 2.7.3); then, we qualitatively compare them in Table 9.

#### 2.7.1. Smart Parking Management (SPM)

The authors of [111] suggested an edge computing-based shared bicycle system, with a hybrid ML model (SOM-RT) and a self-organizing mapping network to assemble the original samples in the form of clusters, and each cluster was built as an RT to forecast the necessary number of bikes at each station. Experiments outperformed other methods in terms of prediction accuracy and generalization.

The authors of [112] developed a camera-based object-detection solution for parking surveillance. They used a single-shot multibox detector (SSD) and background-based detection method in pipeline at the edge to reduce the data transmission volume and ensure efficient updates, whereas the detection results are combined on the server to perform parking occupancy detection in extreme lighting conditions and occlusion conditions with a tracking algorithm for vehicle tracking in parking garages.

In [113], Huang et al. created the fedparking federated learning framework for the management of parked vehicle-assisted edge computing (PVEC). Fedparking uses federated learning with LSTM to estimate parking space. Fedparking enables many parking lot operators to jointly develop a model to forecast the availability of free parking spots in a parking lot in real time for traffic management. For PVEC, they utilized an incentive system. A multiagent deep reinforcement learning strategy was utilized to progressively attain the Stackelberg equilibrium in a distributed yet privacy-preserving way while taking into account the dynamic vehicle arrivals and time-varying parking capacity limitations. High convergence accuracy is obtained by this method.

#### 2.7.2. Traffic Monitoring/Prediction (TMP)

To solve the dynamic traffic changes issue in smart transportation for accurate traffic prediction and for identifying the abnormal situation in real time, the authors of [114] proposed a model for collaborative optimization of intelligent transportation systems. Installing monitoring sites at various traffic crossings allows for data collection from each intersection. The DBN-SVR approach is used to anticipate traffic conditions and predict the overall traffic flow of the road network. Advanced computer technology was employed to process the information signals produced by the crossings after the model was used to determine the traffic flow of a few chosen intersections.

For accurate real-time traffic flow prediction, a framework named AAtt-DHSTNet based on fog computing is proposed in [115]. The authors used an aggregation method based on an attention mechanism to eliminate redundant data acquired by sensors in overlap regions, along with a spatial and temporal correlation-based DHSTNet model, which dynamically manages spatial and temporal correlations through CNN and LSTM models.

For real-time urban traffic prediction, a short-term traffic flow prediction model based on edge computing is introduced in [116]. The authors used a smooth support vector machine optimized by a chaotic particle swarm optimization algorithm.

The authors of [117] proposed a federated learning approach to predict the number of vehicles in an area. First, they used clustering to group participants. Then, they trained a global model for each cluster. They used a joint-announcement protocol in the model aggregation mechanism to reduce the communication overhead of the algorithm.

In [118], the authors proposed an edge computing-based graph representation learning approach for short and long traffic flow prediction. The authors used a federated learning approach. Each model at the edge consists of three components: (1) recurrent long-term capture network (RLCN) module, (2) attentive mechanism federated network (AMFN) module, and (3) semantic capture network (SCN) module for spatiotemporal information in each area. The authors used an additive homomorphic encryption approach based on vertical federated learning (VFL) to share the model.

#### 2.7.3. Intelligent Transportation Management (ITM)

In [121], the authors introduced a system based on edge/cloud computing for real-time driver distraction detection by using a custom DCNN model and a VGG16 (namely, visual geometry group-16)-based model.

A driving behavior evaluation technique built on a vehicle edge-cloud architecture is taken into account by Xu et al. in the work at [119]. When a car is operating on the road, its telematics box transmits data displaying the autopilot/driver behaviors to the edge networks. The driving behavior evaluation model built by the cloud server is used by the edge networks, which then communicate the behavior rankings back to the cars. The driving behavior evaluation model is continually trained and optimized on the cloud server by using vehicle data, and the model is periodically sent to the edge networks for updates. The suggested scheme’s robustness and feasibility are demonstrated by experimental findings.

A methodology for diagnosing railway faults based on edge and cloud collaboration is created in [120]. The model first uses a SAES-DNN for the fault recognition method on the cloud. Then, for a real-time fault diagnosis, a transfer learning strategy is used to assign the task on the edge.

### 2.8. Security and Privacy in Edge-Based Applications

With the recent exponential sophistication of attacks and unauthorized access and in order to ensure and improve the privacy and security of edge-based IoT applications, putting an AI-based solution at the edge of the network is necessary.

In this section, we review and classify AI-based security solutions at the network edge for IoT-based applications into three categories: those that provide early detection of malware and intrusions before the data is delivered to the cloud (Section 2.8.1), unauthorized access solutions (Section 2.8.2), and privacy-preserving solutions to help keep sensitive information safe during data sharing (Section 2.8.3); then, we compare them in Table 10.

#### 2.8.1. Privacy Preservation (PP)

Kumar et al. [122] suggested two techniques for privacy preservation: blockchain and deep learning implemented on the fog nodes in the Collaborative Intelligent Transportation System. The blockchain and the smart contract-based module are used at the first level to support the exchange of nonmutable data. The deep learning module LSTM-AE is used to encode the C-ITS data into a novel format to prevent attacks. Finally, an attention-based RNN is employed for attack detection.

Similar to this, Kumar et al. [123] proposed an integrated safe privacy-preserving architecture for smart agricultural drones that integrates blockchain and DL methods. The framework uses two levels of privacy. A blockchain-based ePoW and smart contracts are included in the first level, and an SAE approach to transform data into a new encrypted format is included in the second level. It uses a stacked short-term memory (SLSTM) anomaly detection engine.

Authors in [124] proposed a model based on differential privacy, called differential privacy fuzzy convolution neural network framework (DP-FCNN). First, they used the addition of noise to protect sensitive information by using a fuzzy CNN with a Laplace mechanism, then secured data storage, and encryption with a lightweight encryption algorithm named PICCOLO before uploading it to the cloud.

To prevent leakage of users’ privacy-sensitive data, authors in [135] proposed a federated learning with a blockchain-based crowdsourcing framework. The authors used differential privacy to protect the privacy of customers’ data. The model updates are accountable for preventing malicious customers or manufacturers from using the blockchain.

#### 2.8.2. Authentication and Authorization (AA)

The authors of [8] presented a DL-based physical layer authentication strategy that takes advantage of channel state information to improve the security of MEC systems by spotting spoofing attacks in wireless networks. The DL-based multiuser authentication method put forward in this research can successfully distinguish between trustworthy edge nodes, malicious edge nodes, and attackers, greatly enhancing the security of MEC systems in the IoT.

In order to achieve high efficiency and the most effective use of computing resources, the study in [125] presents an effective implicit authentication system called edge computing-based mobile device implicit authentication (EDIA). The gait data from the built-in sensors are processed in an optimum manner, and the model is based on the concatenation of CNN and LSTM. By transforming the gait signal into an image, data preprocessing is utilized to extract the characteristics of the signal in a two-dimensional space. A hybrid approach using CNN and LSTM is used for user authentication, with CNN serving as a feature extractor and LSTM serving as a classifier. The technique of authentication also achieves excellent authentication accuracy with modest datasets, demonstrating that the model is appropriate for mobile devices with limited battery and processing resources.

#### 2.8.3. Intrusion Detection (ID)

Samy et al. proposed in [126] a distributed fog framework for IoT cyberattacks by using the LSTM model. First, with the aim of achieving the scalability of the system, a clustering-based mechanism is applied to the fog nodes to balance the network load and increase network scalability and secure the exchanged traffic between the fog and the cloud. The proposed framework has proven its effectiveness in terms of response time with a high detection accuracy compared to cloud-based attack detection systems.

In [127] authors proposed a fog-based framework for the detection of attacks by using a hybrid DL model CNN-LSTM with the use of centralized controller SDN to reduce computation overhead with a highly cost-effective dynamic.

In [128] an IDS is proposed based on the DL approach by using AE and isolation forest (IF) in a fog environment. After identifying the attack and separating it from data from regular network traffic, AE uses an isolation forest to find the outlier data points.

The authors of [129] proposed a lightweight algorithm for resource-constrained mobile devices for attack detection by using a stacked AE, mutual information (MI), and wrapper for feature extraction and SVM for the detection.

In [130], Huong et al. proposed an IoT platform that uses edge and cloud computing for attack detection based on multilayer classification and federated learning. A feature extraction-based PCA coupled with an optimized neural network is implemented for a low-complexity model and good accuracy. However, there is a limitation in the model, which consists of the imbalanced distribution of the data on fog nodes. This limitation decreases the accuracy of detection for some types of cyberattacks.

In [131], Gavel et al. designed a fog-based model for intrusion detection in an IoT network. The model is based on a combination of the Kalman filter and the salp swarm algorithm. First, the Kalman filter is used as a data fusion technique that reduces the redundant data at the fog node. Then, the salp swarm algorithm is used to select the optimum number of features. Finally, the features selected are used to train the model using the kELM classifier. Results achieve highly reduced data, and high detection accuracy with reduced computation time.

An investigator digital forensic algorithm was proposed in [132] to detect and categorize advanced persistent and Shamoon attacks in a fog environment. The model consists of two steps. The first one allows the extraction of the relevant features and the prediction of the best-weighted features with FPSO (frequencies PSO). In the second step, these latter are clustered by using K-means and classified with the KNN.

The authors of [133] introduced a threat detection model at the edge layer based on multikernel SVM. A feature selection module based on GWO is applied to minimize the computational costs of the proposed model by selecting the relevant features. The proposed model achieved high accuracy and outperforms DNN and fuzzy-based IoT malware-hunting techniques. Moreover, it significantly reduces the computational cost and training time.

## 3. Discussions of Related Works: Findings and Insights

In this section, we discuss the works reviewed in Section 2 through different points: (1) the relevance of integrating AI and edge computing in IoT-based applications (Section 3.1); (2) AI technologies (Section 3.2); (3) AI use at the network edge (Section 3.3); (4) enabling technologies and strategies that provide analytic services at the edge (Section 3.4); (5) platforms and software tools (Section 3.5); (6) performance metrics (Section 3.6); and (7) the convergence of AI-edge with other technologies (Section 3.7).

### 3.1. The Relevance of Integrating AI and Edge Computing in IoT-Based Applications

The chart in Figure 2 shows a statistical distribution of the domains considered in this review, which means that smart healthcare is the most studied domain, whereas the distribution in the other domains is almost equal except for smart education, which is the lesser one with 6% of the total number of studies.

From the reviewed works, we have drawn several conclusions considering the benefits of the integration of AI and the edge in the eight reviewed domains (see Table 11).

### 3.2. AI Technologies

Figure 3 shows the classification of the different AI techniques used in the reviewed works. Although Figure 4 shows the percentage of the use of convolutional ML and deep learning algorithms in the reviewed works.

### 3.3. AI Use at the Network Edge

We show in Figure 5 a categorization, which summarizes the use of the AI at the edge of the network. The AI is used for (1) data preprocessing (aggregation, filtering, imputation, and reduction), (2) data analytics (prediction, classification, visualization, and decision-making), (3) resources management (task scheduling, and load balancing), and (4) intelligent sensing (data collection, and data transmission).

### 3.4. Enabling Technologies and Strategies that Provide Analytic Services at the Edge

We conclude from the reviewed works that most of the research studies considered lightweight models with 37% of the total number of reviewed works. The second considered technologies are the transfer learning and federated learning with 25% and 15%, respectively. Whereas, approximately 12% and 9% of the studied related works considered hardware and software optimizations and preprocessing at the edge, respectively. Unfortunately, only 2% of the reviewed works considered DNN splitting and early exit. Figure 6 shows the distribution of these enabling technologies from the reviewed works.

### 3.5. Platforms and Software Tools

For edge-based applications, many simulators are used, such as iFogSim and YAFS. However, for distributed data management at the edge, many big data analytic platforms are used, such as Apache, Spark, and HDFS. Many libraries are proposed for deep learning implementation, such as TensorFlow, Keras, and Caffe. However, with the purpose of enabling deep learning inference at the edge, the lightweight library TensorFlow light is used. Figure 7 shows the platforms used in the reviewed papers.

### 3.6. Performance Metrics

As depicted in Figure 8, the used metrics are low latency, accuracy, training/ inference time, data transmission rate, throughput, stability, mobility, security and privacy, scalability, memory usage, reliability, training time, and bandwidth management. As shown in Figure 8, the most used metrics in the reviewed works are accuracy and low latency. Then, security and privacy, and training time were moderately used. However, a weak use considered the other metrics.

### 3.7. The Convergence of AI-Edge with Other Technologies

Blockchain provides ultrasecurity mechanisms by using cryptographic algorithms [136]. Blockchain is a decentralized ledger system where digital files are grouped into blocks, such as transaction lists or contractual agreements, and stored in a distributed database blockchain smart contract is leveraged to generate a global model by averaging the sum of locally trained models submitted by users. In this federated way, source data are supposed to maintain security and privacy. Due to its distinctive characteristics, such as decentralization, immutability, and traceability, the authors of [137] provide appealing solutions for FL-based intelligent edge computing. FL can be implemented by using decentralized data ledgers rather than a central server, reducing the chance of single-point failures. Any update events and user actions are transparently tracked by all network entities.

## 4. Open Issues and Future Directions

Many factors impact the performance of edge-based smart applications: IoT data quality, *5V* IoT data features, heterogeneity, dynamicity of the edge computing, and its resource constrained. Below, we discuss and present the major issues related to the design and implementation of edge analytics, and we present future directions. As depicted in Figure 9, the major issues revealed from the reviewed works are (1) big data analytic issues (Section 4.1), (2) scalability (Section 4.2), (3) resource management (Section 4.3), (4) security and privacy (Section 4.4), and (5) ultralow latency requirement (Section 4.5).

### 4.1. Big Data Analytic Issues

With the goal of transforming information into actionable insights and retrieving the necessary knowledge for robust decision-making support and a reliable QoS; various issues arise for big IoT data analytics in edge-based applications. The different issues are discussed in detail below:With regard to data quality issues, the collected IoT data may include irrelevant, redundant, and missing data due to IoT network issues such as failure of devices, less coverage, the overlapping area of redundancy that cause high energy consumption and affect the limited power capabilities of IoT devices. All of these features may reduce the accuracy of the model while increasing the execution time and the computational complexity of the analysis. The authors of [54,57,102,138] used AI for spatial and temporal redundancy, data imputation, sensing coverage, and pipeline data preprocessing at the edge, respectively. However, not all of them consider the mobility, dynamic, and heterogeneity feature of an edge environment. The solutions based on (1) dynamic network management, (2) lightweight AI data fusion at the network edge, and (3) quality-aware, energy-efficient data management and data reduction at the network edges are still open issues. AI and 6G/5G are recommended solutions for efficient 3D coverage and intelligent sensing.With regard to analytical learning model choices to deal with IoT big data characteristics, we find the following.Spatio-temporal correlated data issue: Large-scale distributed geographic systems, such as large-scale environmental monitoring and city-wide traffic flow prediction, where data is captured from different geographic locations in continuous time, require the handling of the complex correlation between space–time dependency. Graph-based deep learning is considered a promising solution to handle the spatiotemporal correlation issues [139,140].Nonstationary, dynamic, and nonlinear IoT time series data: It is difficult for classical methods to extract effective features from the collected IoT data due to the nonstationary, dynamic, and nonlinear IoT data, such as in electric power systems. To this end, selecting a suitable model to deal with IoT data characteristics and in order to solve the problems associated with dynamic IoT data, it is desirable to develop an online/incremental learning model that can be further improved to become more flexible and adapt more quickly to changes in the IoT environment. Reservoir computing is used in [40] to deal with this problem. Retraining the deep learning model is still a problem due to the limited recourse constraint of the edge.Generalized, adaptability, and tradeoff between training/inference time and accuracy in ML models are also still challenges to be considered.Limited available dataset, multiclass classification, and imbalanced data set are also challenges to be considered.Frameworks and simulators: To support real-time analysis and development of fog computing, the authors of [141] developed modular simulation models for service migration, dynamic distributed cluster formation, and microservice orchestration for edge/fog computing based on real-world datasets. In [142], the authors proposed a multilayer fog deployment framework for job scheduling and big data processing in an industrial environment.With regard to device computation, we find the following.Hardware and software optimization challenges: In the literature, many hardware platforms capable of accelerating DL execution are used like server-class central processing units (CPUs), and graphics processing units (GPUs). As an innovative solution and to enhance the efficiency of computing in edge devices. Hardware implementation is designed as an integrated solution to the neural network in [143].Model compression challenges: Many solutions emphasize employing quantization and compression methods to address the limited hardware requirements of an edge device and compress CNN. The quantization requires careful tuning or retraining of the model, which can take a long time and affect the accuracy of the model. Other solutions use dynamic compression with an effort to reduce model complexity and eliminate redundant components, such as in [56]. Others formulate CNN model compression as a multiobjective optimization problem with three functional objectives: reducing the size, improving classification accuracy of the DCNN, which is related to the reliability of the model, and minimizing the number of neurons in the hidden layer using the Lévy flight optimization algorithm (LFOA) [59]. This model suffers from high complexity in training time. One of the future directions could be the combination of dynamic compression with quantization for more accuracy [56].With regard to distributed and parallel computing, we find the following.Federated learning:−Communication overhead: FL involves sharing the model parameters instead of the data. Transmitting complex models from large numbers of clients to centralized aggregators generates a massive load of traffic, which makes communication overhead. The iterative and nonoptimized methods of communication between the server and the clients are the main factors for increasing the communication overhead. Decreasing the communication frequency at each round is also essential to improve the efficiency of the algorithm considering the bandwidth cost. As a solution, authors in [144] proposed federated particle swarm optimization (FedPSO) for transmitting score values instead of large weights, which reduces the overall traffic in the network communication. Moreover, authors in [145] proposed a framework called COMET, in which clients can use heterogeneous models. It uses knowledge distillation to transfer its knowledge to other customers with similar data distributions.−Fault tolerance: Reliability and fault tolerance means the whole system architecture should be able to provide services even if any node (server) on any level fails [146]. Leveraging peer-to-peer FL updates model in the coordination of training can eliminate the single point of failure that may be inherent in an aggregator-based approach [33]. Authors in [147] proposed a decentralized learning variant of the P2P gossip averaging method with batch normalization (BN), adaptation for P2P architectures. BN layers accelerate the convergence of the nondistributed deep learning models.−The unbalanced and not independent and identically distributed (Non-IID) data: Non-IID data on the local devices (divergence in the data distribution) can significantly decrease learning performance. Many solutions proposed to solve this problem, such as model selection, and clustering are reported in [20,116].With regard to DNN splitting, its advantage is that, compared with model compression, it will not lose accuracy. However, it will create many caching and communication costs because tasks should be transferred between the edge nodes to reach the appropriate nodes with low delay and sufficient resources [148]. Early exit is used by [76] to overcome the limitation, but choosing the point of early exit is still inconvenient. Other problems are related to heterogeneous node failure, and many solutions in the literature are proposed, such as RoofSplit [148], which is used to overcome the limitation of communication cost. SplitPlace is used for mobility. Therefore, developing a heterogeneous, parallel, and collaborative architecture for edge data processing for various DL services will be helpful. Other solutions still need to be developed.

### 4.2. Scalability

Edge computing has a scalability problem when high-volume IoT devices require processing at the edge. Inadequate distribution of computation across multiple resource-constrained nodes affects the scalability of the system. In the works reviewed above, few works considered the scalability problem in edge-based applications. For example, in Samy et al. [126], clustering of fog nodes to balance the network load is used to increase scalability. Autoscaling is a solution that aims to optimize the use of resources [149]. However, the edge-computing environment is very dynamic which impacts the availability of nodes in a distributed edge-based infrastructure, so the load on each node may change continuously. Therefore, the scaling of processing services must be dynamic. Recent work has studied online machine learning for autoscaling, such as the one in [150], in which the authors present an autoscaling subsystem for container-based processing services. However, it will be interesting and promising to design dynamic autoscaling to ensure the scalability of the system with high QoS performance.

### 4.3. Resource Management

Edge computing is a resource constraint. Task scheduling and load-balancing tasks across fog nodes are crucial to improve the quality of service of IoT-based applications, including response time and improving the usage of fog nodes. In the distributed architecture of edge-based applications, different edge servers or fog nodes are shared to perform the processing of the collected data. The load imbalance among the edge servers affects the stability of the system. Many works, such as [109], propose dealing with resource-management issues. However, none of them consider heterogeneous and dynamic node distribution. Dynamic load balance is an efficient solution, such as in the study in [151], in which authors proposed a network traffic-based dynamic load balancing approach to optimize the overall network performance.

### 4.4. Security and Privacy

The security problems of edge nodes are more important than those of servers because they are placed at the edge of the networks, closer to the attackers. Therefore, an authentication security mechanism must be developed. The use of machine learning in adding noise for differential privacy is a promising solution for improving the security and processing time of the system. For example, in the reviewed works, DL-based physical layer authentication approaches can distinguish multiple legitimate edge nodes from malicious nodes and attackers. Moreover, DL is used for encoding data into a new format that prevents inference attacks from gaining knowledge relative to original datasets.

### 4.5. Ultralow Latency Requirement

The need for ultralow service requires to introduce tactile 5G [152]. For example, authors in [153] proposed a solution for ultralow latency based on machine learning and network slicing.

## 5. Conclusions

This paper attempts to provide a review of edge computing-based applications with a focus on the fusion of AI and edge computing while offering discussions on future research directions related to AI and edge computing convergence. We started with a review of existing recent works in eight different IoT-based application areas, and we qualitatively compared them through tables by using several characteristics (use case, reference, contribution, AI role at the edge, AI algorithm, dataset, AI placement, employed technology, platform, metrics, benefits AI-Edge, and drawbacks). Then, we discussed the related works to distinguish what was already done and used for the convergence of AI and edge. After that, we presented issues and open challenges that serve as guidelines for future work.

This review is limited to aspects related to the confluence of AI and Edge in eight application areas from a global perspective for the purpose of big data analytics at the edge. In this sense, this article focuses only on papers that deal with edge learning in distributed edge-based architecture. It only touches on task and resource management and the different feature challenges of edge in a limited way.

## Figures and Tables

**Figure 1 sensors-23-01639-f001:**
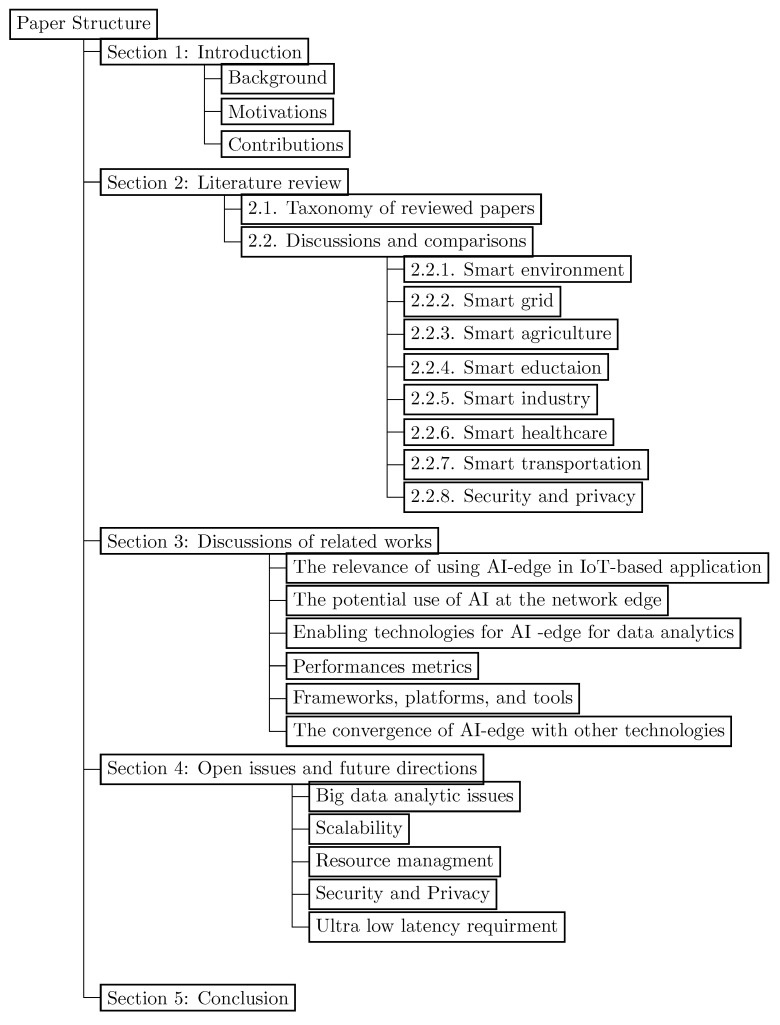
A schematic overview of the paper organization structure.

**Figure 2 sensors-23-01639-f002:**
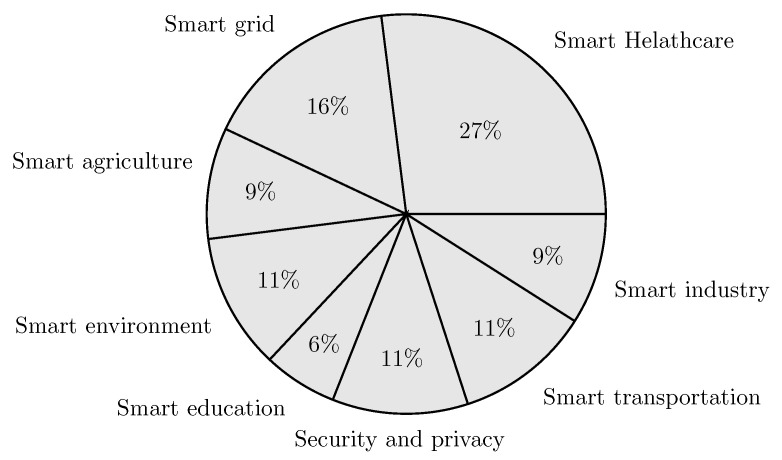
A statistic distribution of the domains considered in this review.

**Figure 3 sensors-23-01639-f003:**
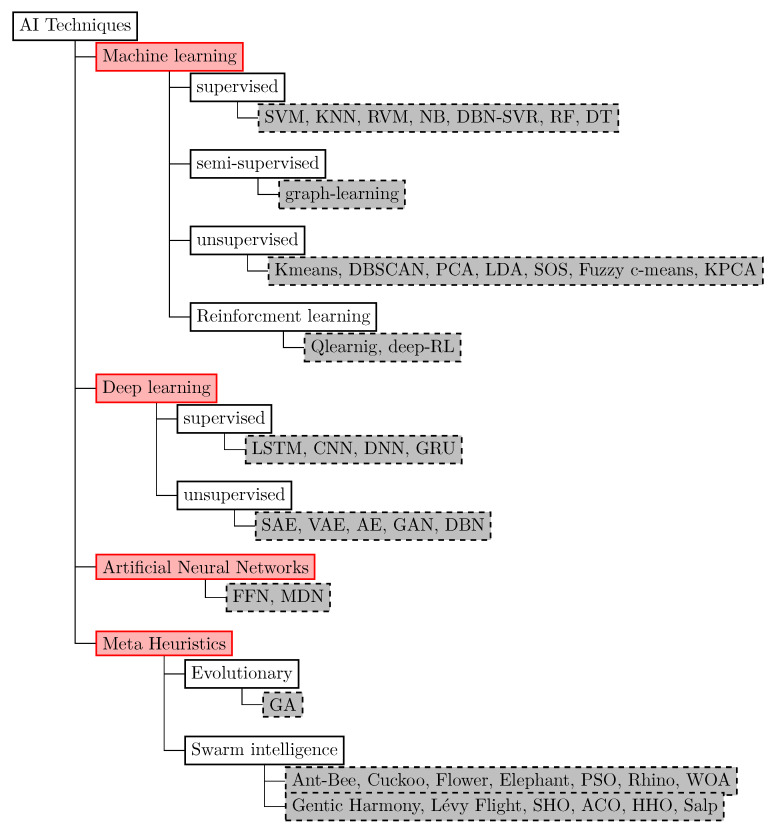
Categorization of AI technologies.

**Figure 4 sensors-23-01639-f004:**
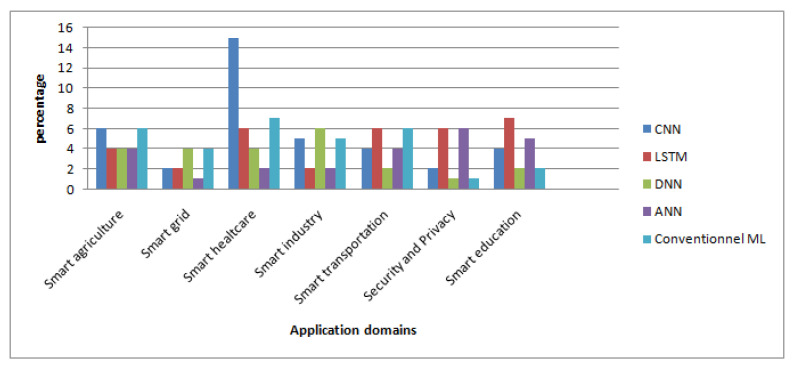
Percentage of the use of the convolutional ML and deep learning algorithms in the reviewed works.

**Figure 5 sensors-23-01639-f005:**
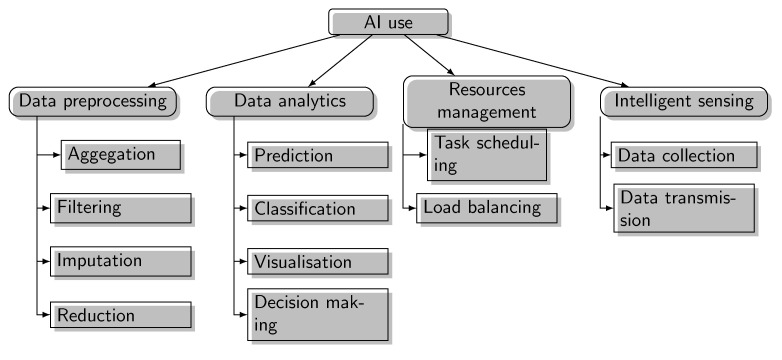
AI use.

**Figure 6 sensors-23-01639-f006:**
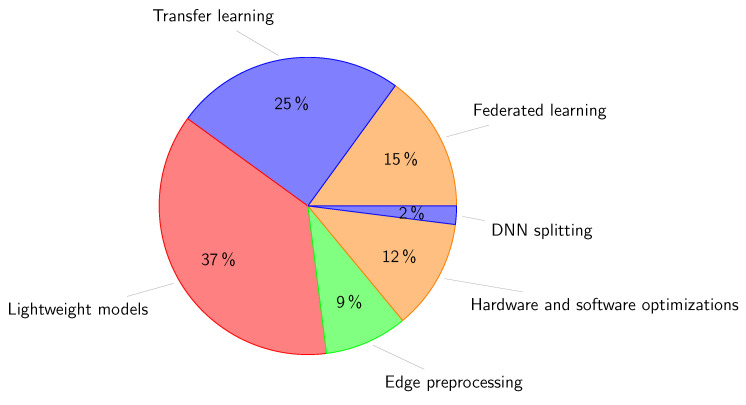
Enabling technologies.

**Figure 7 sensors-23-01639-f007:**
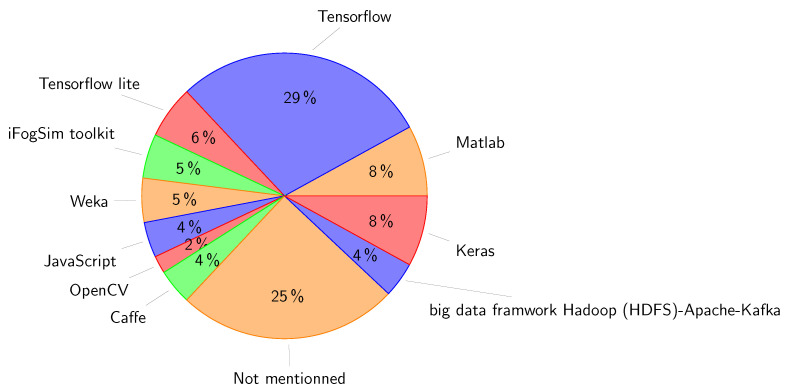
Platforms and software tools.

**Figure 8 sensors-23-01639-f008:**
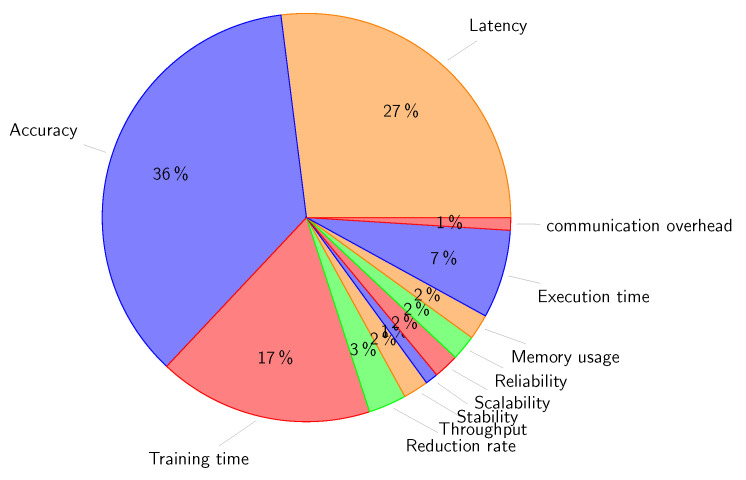
Performance metrics.

**Figure 9 sensors-23-01639-f009:**
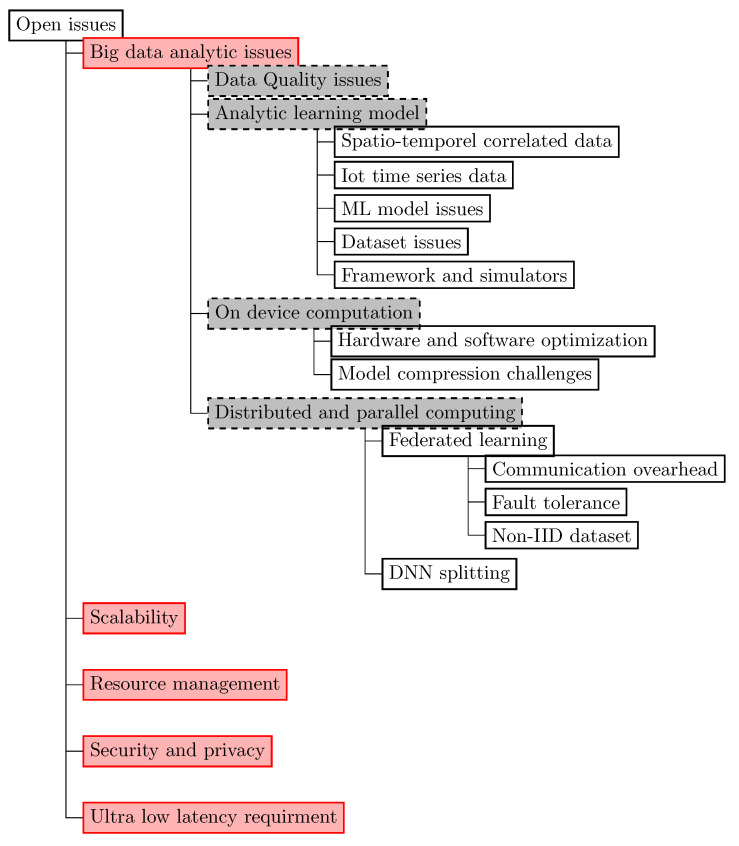
Open issues.

**Table 1 sensors-23-01639-t001:** Qualitative comparison of related works.

Year	Reference	AI Category	Big Data Analytics	Resource Management	Key Enabling Technologies	Application Domains
2021	[14]	No	No	Yes	No	Yes
2022	[13]	Yes	No	No	No	IoV
2021	[3]	Yes	No	Yes	Yes	yes
2020	[15]	yes	No	Yes	Yes	NO
2020	[16]	Yes	Yes	Yes	Yes	No
2019	[17]	Yes	Yes	Yes	No	Yes
2020	[19]	Yes	No	Yes	No	No
2022	[18]	Yes	No	Yes	No	Yes
2023	Our paper	Yes	Yes	Yes	Yes	Yes

**Table 2 sensors-23-01639-t002:** Taxonomy of the reviewed works.

AI-edge based applications	Smart environment	AQM	[20,21,22,23,24]
		WQM	[25,26,27]
		SWM	[28]
		UM	[29,30,31,32]
	Smart grid	LDF	[33,34,35,36,37,38,39,40]
		DSM	[41,42,43]
		LAD	[44,45,46,47,48,49]
	Smart agriculture	WP	[50,51]
		LM	[52,53]
		SI	[54,55]
		CMDD	[56,57,58]
		MHSAM	[59,60]
	Smart education	SEM	[61,62,63,64,65]
		SA	[66,67,68]
	Smart industry	FI	[69,70]
		CI	[71]
		MMM	[72,73,74,75,76]
		PQMP	[77,78,79]
	Smart healthcare	DHM	[80,81]
		AAL	[82,83,84,85,86]
		HAR	[87,88,89,90,91,92]
		LDP	[93,94,95,96,97,98,99,100]
		DD	[101,102,103,104,105,106,107,108,109,110]
	Smart transport	SPM	[111,112,113]
		TMP	[114,115,116,117,118]
		ITM	[119,120,121]
	Security and privacy	PP	[122,123,124]
		AA	[8,125]
		ID	[126,127,128,129,130,131,132,133]

**Table 3 sensors-23-01639-t003:** Qualitative comparison of smart environment-related works.

	Use Case	Ref	Contribution	AI Role (At the Edge)	AI Algorithm	Dataset	AI Placement	Employed Technology	Platform	Metrics	Benefits AI-Edge	Drawbacks
Smart environment	AQM	[20]	Predicting of futureindoor status of PM10 and PM2.5	Prediction	LSTM	Data from Seoul, Korea	Edge device, cloud	Federated learning	TensorFlowKeras	RMSE	Minimize load Hight accuracy	Does not consider all factors in prediction
		[21]	Green energy-based wireless sensing network for air-quality monitoring	Prediction	LSTM	Airbox system dataset	Edge device, cloud	Federated learning	Not mentioned	MAE-loss RMSEEnergy thresholdsaving, ratio error rate	Communication efficiencyPreserving data privacy Low computational complexity	Slightly lower accuracy
		[22]	Location awareenvironment sensing	Prediction	k-means, LSTM, CNN (ResNet)	WA dataset Outdoor image datasets	Edge device, cloud	Distributedcomputing cluster	Federated learning	Accuracy, avg. sum of squared errors, silhouette coefficient	High accuracy	Homogeneous nodes only considered
		[23]	Distributed data analysis for air prediction	Preprocessing	K-means SVM, MLP, DT, KNN, NB	U.S. Pollution Data Kaggle	Edge devices, cloud	Distributed computing	IFogSim toolkit-YAFS-	AccuracyPrecision recallF1-Score	Data reductionLow response time reduction	Not consider mobility of nodes
		[24]	On-device air-quality prediction	Prediction	CNN, LSTM	Dataset from University of California–Irvine (UCI) Machine Learning Repository page	Edge devices(RPi3B+, RPi4B)	Posttraining quantization Hardware accelerator	TensorFlow Lite	RMSE, MAEexecution time	Low-complexity model latency	Accuracy degradation
	WQM	[25]	Onboard sensor classifier for the detection of contaminants in water	Classification	EA PCA	Real-world dataset	Edge device (sensors)	Low-cost model	Not mentioned	Accuracy F-score TP TN FP FN	High accuracy	Low accuracy for unlabeled data
		[27]	Online water-quality monitoring	Prediction	BPNN	Real-world dataset	Edge gateway	Low-cost model	Not mentioned	Data transmission response time	Low-complexity model accuracy, data transmission reduction	Accuracy needs to improved
	WQM	[26]	Real-time water- quality monitoring	Preprocessing prediction	PCA LR MLP SVM SMO Lazy-IBK, KStar RF RT	Data of sewage water-treatment plant of the institute, data collected from river Ganga	Edge device (Raspberry Pi)	Transfer learning	Python, Weka	Correlation coefficient MAE RMSE-RAE RRSE Edge response time	Less response time	Communication cost not considered
	SWM	[28]	Smart water saving and distribution	PredictionDecision making	FFN MDN	Real-world dataset	Edge server	SofT computing blockchain	Python	MSE accuracy	Effective decision-making	Accuracy needs to be enhanced
Smart environment	UM	[29]	Reduce data and improve data quality or underwater	Data (fusion, reduction)	BPNN evidence theory	Western Pacific measurement information	Fog gateway Cloud	Edge preprocessing	Not mentioned	Time consumption Redundant data volume R, MAE, MSE SMAPE	Low communication costHigh accuracy	High delay
		[30]	Real anomaly detection errors in underwater vehicles	Network management, data reduction classification, decision-making	YULO (CNN), RL	Real-world dataset	Edge device (Raspberry Pi) Fog gateway	Hardware accelerator, pretrained CNN	Not mentioned	Accuracy, latency, recall	High accuracy, less latency	Accuracy degraded
		[31]	Low delay for Seawater quality prediction	Data reductionPrediction	PCA RVM	Real-world dataset	Mobile edge computing	Low-cost model	Not mentioned	CD MAE RMSE	Higher prediction Low time consumption	High-cost model
		[32]	Downlink throughput performance enhancement	Resource allocation Classification	DRL DNN	Real-world dataset	Edge device (IoUT devices)	Federated learning	Not mentioned	Downlink throughput channel usage Convergence rate	Low complexity	–

**Table 5 sensors-23-01639-t005:** Qualitative comparison of smart agriculture related works.

	Use Case	Ref	Contribution	AI Role (At the Edge)	AI Algorithm	Dataset	AI Placement	Employed Technology	Platform	Metrics	Benefits AI-Edge	Drawbacks
Smart agriculture	WP	[50]	Timely prediction of frost in crops	Prediction	LSTM	Real-world dataset	Edge device (Nvidia Jetson)	Hardware accelerator	TensorFlow 1.10.1 Keras 2.2.4	Power consumption, execution time, RMSE, MAE, memory usage, PCC R2	Less execution time	Less scalability complexity of model causes overlearning and slightly increased error
		[51]	Drought prediction	Feature extraction	ANN, PCA, GA	Drought attribute dataset	Fog gateway, cloud	Preprocessing edge	Matlab Amazon EC2	Accuracy sensitivity specificity, precision, F-measure	Reduction of load to cloud High accuracy	High Communication cost
	LM	[52]	Livestock surveillance	Feature extraction	CNN	Google ImageNet Pixabay	Edge device (Nvidia Tegra) Cloud	Splitting DNN	Caffe	Accuracy Reduction rate	Load reduction High accuracy	High communication cost
		[53]	Early lameness detection in dairy cattle	Feature extraction	K-means, KNN	Real-world dataset	Fog gateway (PC), cloud	Edge preprocessing	Python	Reduction rate Accuracy	High accuracy	High communication cost
	SI	[54]	Prediction models of soil moisture	Missing-data imputation, prediction	GDR, LSTM, BiLSTM	Coconut, Cashew datasets	Single-board computer (Raspberry Pi 4 Model B)	Hardware accelerator	TensorFlow	CPU RAM usage, MAE	Data quality improvement High accuracy	Accuracy must be improved
		[55]	Intelligent irrigation system	Prediction	LSTM GRU	Historical Hourly Weather Data 2012–2017	Edge devices	Hardware accelerator/software	Pytorch, TensorFlow, TensorFlow Lite	RMSE, MSE, MAE	Reliability	Overhead computation
	CMDD	[56]	Timely diagnosis of crop disease	Prediction	CNN	Real-world dataset	Edge device (STM32F746G-disco board)	Quantization	TensorFlow Lite	Accuracy, memory usage, inference time, energy consumption	High accuracy Low memory usage	Accuracy may degrade
		[57]	Timely recognition of crop Diseases	Classification	CNN	Real-world dataset	Mobile edge device	Transfer learning	Python	Accuracy	High accuracy Less recognition time	High computational cost
	CMDD	[58]	Intelligent sensing in the entire crop life cycle	Preprocessing network management	Fuzzy Gath–Geva clustering, Tkagi–Sugneo-fuzzy neural network, KNN, BPNN	Real-world dataset	Edge server	–	Not mentioned	AFE CC accuracy Sensing time, communication rate	Data collection times reduction Less energy consumption Sensed data quality improvement High accuracy	–
	MHSM	[59]	Timely vehicle health monitoring	Prediction	ANN GA	Not mentioned	Smartphone	Lightweight model	MATLAB 2019b	Accuracy, ROC curve, misclassification rate, MSE	High accuracy	Complexity reduction still recommended
		[60]	Vehicle health recognition	Classification	DCNN Levy flight	Real-world dataset	Smartphone	Lightweight DL	Not mentioned	Accuracy ROC, precision recall, F1-score	Low complexity	High training time

**Table 6 sensors-23-01639-t006:** Qualitative comparison of smart education related works.

	Use Case	Ref	Contribution	AI Role (At the Edge)	AI Algorithm	Dataset	AI Placement	Employed Technology	Platform	Metrics	Benefits AI-Edge	Drawbacks
Smart education	S. engagement monitoring	[61]	Attention detection of participants	CNN	Prediction	(DAiSEE)	Edge ( pc)	Pretrained model	Python	Accuracy	-	Accuracy needs to improve
		[62]	Improve long-distance education	Classification	ResNet-50	Fer2013 emotion dataset	Mobile edge computing	Hardware accelerator	/	Confusion matrix accuracy	High accuracy	Accuracy needs to improve
		[63]	Real-time intervention in negative emotional contagion in a smart classroom	Classification	CNN	Fer2013 emotion dataset	Edge preprocessing	Hardware accelerator	JavaScript, TensorFlow, OpenCV	Accuracy	Less response time	Accuracy needs to improve
		[64]	Multimodal engagement analysis	Prediction	DL	Real-world data	Edge server (PC)	/	JIFF, JavaScript library, TensorFlow	Average performance impact on edge device /server	Scalability	Computational overhead
		[65]	Student stress monitoring and real-time alert generating	Prediction	VGG16, BiLSTM, NB	Real-world data Kaggle dataset	Fog cloud	Cloud training	Not mentioned	Specificity, sensitivity, accuracy, F-measure	High accuracy	Eliminate historical record
	Skill assessment	[66]	Monitors the academic/skill of students for timely employability classification of graduation.	Resource management	K-means, PCA, KNN	Real-world dataset	Fog nodes	/	iFogSim toolkit	Mean absolute percentage error (MAPE)	Scalability	Processing overhead
		[67]	Education quality evaluation		ANFIS Bayesian belief network (BBN)	Environmental datasets, staff-related dataset, physical dataset, students’ academic-related historical dataset	Raspberry Pi v3 is	/	Weka	Precision, specificity, sensitivity, BBM, accuracy, RMSE, MAS	Stability, reliability	Accuracy needs to be improved
		[68]	Ideology and politics education evaluation in 5G	Resource management data caching	PSO	Edge devices	Not mentioned	-	/	Energy consumption, latency	Scalability, low energy consumption, low latency	-

**Table 7 sensors-23-01639-t007:** Qualitative comparison of smart industry related works.

	Use Case	Ref	Contribution	AI Role (At the Edge)	AI Algorithm	Dataset	AI Placement	Employed Technology	Platform	Metrics	Benefits AI-Edge	Drawbacks
Smart Industry	FI	[69]	Financial data analysis	Prediction	SVM	(Credit card fraud, credit card risk, Customer Churn, Insurance Claim) dataset	Edge devices, cloud	Low-cost model task offloading	Simulator (Not mentioned )	Task assignment over delay power consumption precision recall F1-score	High accuracy	Communication overhead
		[70]	Early-warning of financial risks	Prediction	BPNN	Real-world dataset	MEC	Quantization HARDWARE-CPU	Matlab	Accuracy, hit rate	Less response time	Accuracy needs improvement
	C.I	[71]	Locality-based product demand prediction and decision making	Feature selection, classification, decision-making	RL, PCA, K-means	Kaggle open data	Edge device (GPU NVIDIA-SMI)	Low-cost model	Scikit-learn Python	Clustering score maximum/average cumulative reward execution time	Outperform others existing methods	Stability not tested
	MMM	[72]	Machine malfunction monitoring	RF SVM Adab LR MlP	(MIMII dataset	Fog (controller unit (ICU)/Microdata center)	Hardware accelerator	Lightweight model	Not mentioned	Time complexity, accuracy, precision, FScore	Response time reduction	–
		[73]	Abnormal events detection during assembly line production	Outlier detection prediction	RF, DBSCAN	Real-world dataset	Edge devices (Raspberry Pi)	Low-cost model	MongoDB Python	Accuracy recall F1-score precision	High accuracy	Dynamic of IoT data not addressed
		[74]	Fault detection in a hydraulic system	Data reduction classification	LSTM, AE, GA	Real-world dataset	Edge server	Transfer learning	TensorFlow	Complexity DL accuracy detection time, data reduction	Reduction of load to cloud Low detection time Robust to noisy data	Communication overhead
Smart Industry	MM	[75]	Faults of machine detection	Classification	LSTM	Real-world dataset	Edge device (Raspberry Pi)	Lightweight model	Keras Python	Accuracy	Low-cost model Short fault detection	Memory usage overhead
		[76]	Fast manufacture inspection	Feature extraction classification	CNN	Real-world dataset	Fog gateway	Early exit-DNN splitting	Not mentioned	ROC curve running efficiency	High accuracy	High communication cost
	PQMP	[77]	Fast prediction of assembly quality	Feature selection, prediction	RF Adaboost	Real-world dataset	Edge server (PC)	Transfer learning	Python	Accuracy	Efficacy flexibility complexity reduction	Online learning not improved
		[78]	Fast tool wear monitoring and prediction	Feature extraction classification	CNN LSTM BiLSTM	Real-world dataset	Edge server (PC)	Transfer learning	Python TensorFlow	Response time, network bandwidth, data transmission RMSE MAPE	High monitoring accuracy, low-cost model, low response latency	Accuracy loss
		[79]	Scheduling tasks production for smart production line	Task scheduling, resource allocation	PSO, ACO	Not mentioned	Fog gateway	-	Matlab	Completion time, energy consumption, reliability	Solves the problem of limited computing resources, high energy consumption, real-time/efficient processing	Does not consider heterogeneity of IoT devices.

**Table 8 sensors-23-01639-t008:** Qualitative comparison of smart healthcare-related works.

	Use Case	Ref	Contribution	AI Role (At the Edge)	AI Algorithm	Dataset	AI Placement	Employed Technology	Platform	Metrics	Benefits AI-Edge	Drawbacks
Smart healthcare	DHM	[80]	Food recognition	Classification Storage	DRCNN	Food 101Image	Smartphone	Quantization, GPU accelerator	TensorFlow Lite	Accuracy loss values, computational power	Low response time	Loss of accuracy over time
		[81]	Food recognition	Classification preprocessing	GoogLeNet	UEC-256 UEC-100 Food-101	Smartphone	Pretrained CNN	Caffe	Response time, accuracy, computational power	Low response time	Loss of accuracy over time
	AAL	[82]	Accurate and timely fall detection	Classification	LSTM/GRU	SisFall dataset	IoT, gateway (fog)	Virtualization	Docker HDFS-Apache Kafka-MongoDB Tensorflow	Accuracy, sensitivity, precision, inference	Scalability, flexibility	Memory consumption needs to be optimized Mobility not considered
		[83]	Online/offline monitoring elderly patients suffering from chronic disease	Prediction	NB-FA	Vital signs, behavioral data environmental data	Cloud, edge	Transfer learning	Weka, classifier, Spark job	Accuracy, sensitivity, precision, inference time	Accurate, fault-tolerant, fast decisions	High computational cost
		[84]	Real-time fall detection	Preprocessing, prediction	LDA KNN SVM	SisFall datasets	Raspberry Pi 3 B +	Real-time test	Low-cost model	Response time	High accuracy, low response time	Accuracy and generalization still improved
		[85]	Multimodal fall detection	Prediction	PCA linear regression MLP	SisFall data set	Mist, fog, cloud, edge	Not mentioned	Low-cost model	CC, MAE RMSE, RAE, RRSE response time	High accuracy, less inference time	Generalization needs to be solved
		[86]	Real-time in-home health monitoring	Prediction	GCAE	MobiAct dataset	Cloud, edge	Federated learning	Not mentioned	Accuracy communication rounds scalability	Heterogeneity of data and communication cost solved	Data privacy issues
Smart healthcare	HAR	[87]	Real-time abnormal human activities	Prediction	PCA -CNN	UniMiB DATASET	Edge device	Transfer learning	Python 3.6	Process time	Low energy consumption, less computational cost	Lack of security
		[88]	Real-time, human activity recognition	Prediction	DRNN	WISDM dataset	Raspberry Pi3 (edge devices)	Virtualization	TensorFlow	Accuracy F1-score recognition time	Less recognition time, high accuracy	High computational cost
		[89]	Energy-efficient, human-activity recognition	Training, prediction	CNN	Opportunity dataset, w-HAR dataset	Edge devices	Transfer learning	Not mentioned	Accuracy, precision, recall, weighted, F1-score	Less memory overhead, high accuracy	Stability not tested
		[90]	Human activity recognition	classification	SVM	KTH Dataset Hollywood2 Action Dataset	Edge/cloud	Transfer learning Blockchain	TensorFlow	Accuracy	High accuracy multiclass classification	Less scalability
		[91]	Multiaccess physical monitoring system	Classification	BDN	Real-world dataset	Wearable IoT	Transfer learning	Not mentioned	Accuracy data transmission time RMSE	Less energy consumption, high accuracy	Lack of data privacy, less scalability
		[92]	Physical instance-based irregularity recognition	Classification	CNN LSTM	NTU RGB dataset	Fog nodes	Transfer learning	Python-Pillow, OpenCV, Numpy libraries	Rate of latency analysis	High accuracy, less latency	Environmental changes and model generalization not considered
Smart healthcare	LDP	[93]	Monitoring and predicting COVID-19 outspread	Prediction visualisation	FCM T-RNN SOM	-	Fog nodes	MATLAB-Ifogsim	Preprocessing	Latency time, response delay, accuracy, precision	reliability, high accuracy	Lack of security
		[94]	Location-aware monitoring and preventing encephalitis	Prediction visualisation	FCM- T-RNN, SOM	Cloud, edge	UCI-repository data	Preprocessing	MATLAB	Latency time, response delay, accuracy, precision	Reliability, high accuracy, location aware, data management	Lack of security
		[95]	Early detection of Kyasanur forest disease and control the disease outbreak	Classification	ANN	KFD dataset	Fog/cloud	Lightweight model	Not mentioned	Accuracy, sensitivity, specificity, RMSE MAE	High accuracy	High computational cost
		[96]	Continuous monitoring and early detection of mosquito-borne disease	Classification	FNN, SNA graph	UCI-repository data	Fog node	Lightweight model	Not mentioned	Accuracy, sensitivity, specificity	High accuracy	Data integrity and security not considered
		[97]	Automatic diagnosis of COVID-19	Classification	K-MEANS -VGG16	X-ray ultrasound datasets	Edge devices	Pretrained model	TensorFlow	RMSE, MAE	Cope with data heterogeneity	Less accuracy, lack of security
		[99]	Remote COVID-19 diagnosis	classification	RF GAN GNB	Generated dataset	Fog nodes	Open-source language R iFogSim	Accuracy response time, recall	High accuracy	High energy consumption, lack of security	
Smart healthcare	LDP	[99]	Remote COVID-19 diagnosis	Classification	Mobile-Net V2	Chest CT scan image dataset	Transfer learning	Edge devices	TensorFlow	Sensitivity specificity precision F1-score	High accuracy, less response	Not tested for large datasets, accuracy needs to be improved
		[100]	Low delay in prediction of health status of COVID-19 patients	Preprocessing prediction	eRF	COVID-19 dataset	Edge devices	Lightweight model	TensorFlow	Training time, accuracy, precision, recall, MAE, RMSE	High accuracy	High computational cost
	DD	[101]	Early lung cancer diagnosis	Preprocessing, feature selection Classification	FCM, CS, SVM	(ELCAP) dataset	Fog nodes	Lightweight model	MATLAB 2013a	Accuracy, sensitivity, specificity, MCC, F-measure, ROC curves, computational cost	Less training time, high accuracy	High cost of model for fog implementation
		[102]	Intelligent monitoring of cardiomyopathy patients	Intelligent sensing	FHHO, FL	Real-world dataset	Fog nodes	–	Not mentioned	Execution time, accuracy, precision, recall, F-measure	High accuracy, low time cost	Lack of security, high energy consumption
		[103]	Real-time monitoring patients with chronic diseases	Classification	NB-WOA	Clinical dataset, Physio Bank-MIMIC II database	Fog nodes, cloud	Transfer learning	Weka, Spark	Accuracy, recall, precision	Higher accuracy, high response time	High complexity of model, lack of security
		[104]	Early heart disease prediction	data fusion prediction	CFS, KRF	UCI repository data	Fog nodes	Lightweight model	–	Accuracy, training time, scalability	Scalability, accuracy	Quality of the data depends on the number of sensors, improved accuracy is required
Smart healthcare	DD	[105]	Early detection of Parkinson’s Disease	Prediction	ANFIS GWO PSO	UCI University of California	Fog nodes	Distributed computing	TensorFlow	RMSE, MAE	High accuracy	Lack of security
		[106]	Diabetic cardio disease prediction	Prediction	Rule-based clustering, CRA, ANFIS	(Heart disease, diabetes) dataset	Edge devices	Blockchain	Java	Purity NMI accuracy execution time	Efficient grouping medical data, high accuracy, secure data sharing, good training with uncertainty	Low accuracy
		[107]	Remote cardiac patient monitoring	Classification	1D-CNN	MIT-BIH Arrhythmia	Fog nodes (single-board computer), cloud	Transfer learning	Not mentioned	RMSE MAE CPU usage accuracy loss recall precision F1-score	High accuracy, low computational overhead, low resource usage, low response time	Scalability not considered
		[108]	Timely disease diagnosis of health conditions	Data preprocessing classification	AE HMWWO	UCI-repository data	Edge devices	Lightweight model	Not mentioned	Latency, F-measure time complexity sensitivity	High sensitivity, improved accuracy Minimum time complexity and latency scalability	Small dataset used for evaluation, lack of data protection
		[109]	Real-time physiological parameter detection	Preprocessing prediction, load balancing	RK-PCA HMM MoSHO SpikQ-Net	UCI repository data	Edge devices, fog nodes	Lightweight model	iFogSim	Execution, time accuracy, latency	Stability, scalability, low execution, time, low latency, low complexity	Lack of security
		[110]	Real-time blood glucose	Prediction	GRU	(OhioT1DM ABC4D ARISES) datasets	Edge device (Smartphone)	Hardware accelerator	TensorFlow Lite	RMSE, MSE	Low energy consumption, good training with uncertainty	Less sensitivity

**Table 9 sensors-23-01639-t009:** Qualitative comparison of smart transportation-related works.

	Use Case	Ref	Contribution	AI Role (At the Edge)	AI Algorithm	Dataset	AI Placement	Employed Technology	Platform	Metrics	Benefits AI-Edge	Drawbacks
Smart transportation	SPM	[111]	Real-time prediction Bike charging at each stationReduce load to cloud	Prediction	RT SOM	Kaggle competition, London shared bike data	MEC	Lightweight model (ML)	Not mentioned	RMSE RMSLE	High accuracy Generalization	Multivariate data not supported security
		[112]	Real-time parking occupancy surveillance Reduce load to cloud	Classification	Mobile-net SSD, BG, SORT	MIO-TCD	Edge device Raspberry Pi 3B,	Transfer learning	TensorFlow Lite	Accuracy	Flexibility Reliability Online and high accuracy	Accuracy needs to be enhanced (=95),security
		[113]	Privacy preserving Parking space estimation	Prediction, decision making	LSTM DRL Game theory	Birmingham parking dataset	Fog nodes	Federated learning	Not mentioned	MSE	Computation offloading in nonstatic environment, improve security, flexibility, high accuracy	Less convergence speed
	T.M.P	[115]	Timely citywide traffic prediction, context data management	Data aggregation	CNN, LTSM	Beijing taxicabs data NYC bike data	Fog nodes	Transfer learning	IFogSim	Complexity, training time, prediction time, accuracy	Reduce network congestion,increase energy efficiency, less training/prediction times	Cloud inference, non-real-time prediction
		[114]	Forecast the overall traffic, adjust the redirected flow	Prediction	DBN-SVR	Caltrans PeMS	Fog nodes	/	MATLAB	Scalability, processing time, accuracy	Scalability, security	Accuracy needs to be enhanced
		[116]	Privacy preservation Traffic flow prediction	Prediction	GRU, k-means	PeMS database	Edge nodes	Federated learning	Not mentioned	MAE, MSE, RMSE, MAPE	Low communication overheadStatistical heterogeneity solved, high accuracy	Spatiotemporal correlation not solved
		[117]	Timely traffic flow prediction	Prediction	SVM PSO	Guiyang City dataset	Fog nodes	Lightweight ML	Matlab 2014a	MSE	Low time overhead, faster processing, adaptability, good prediction	Model complexity high
		[118]	Spatial traffic flow prediction	Prediction	GCNs	TaxiBJ TaxiNYC dataset	Edge nodes	Federated learning	Not mentioned	RMSE, MSE, MAPE	High accuracy	Less scalability
	ITM	[88]	Driver distraction identification	prediction	VGG1-CNN -k-means	Kaggle’s state farm, distracted driver challenge	Edge deviceRaspberry Pi	Transfer learning	KERAS	Accuracy, precision, recall, F1-score	High accuracy	Securityless scalability
		[119]	Driving behavior evaluation	Prediction	CNN-LSTM	ToN UCI knowledge discovery, archive database	Fog nodes	Transfer learning	TensorFlow	Accuracy-loss curves	High accuracy, generalization	Less scalability, security
		[120]	Real-time fault diagnosis	Prediction	SAES-DNN, knowledge graphs	ToN UCI knowledge, discovery archive database	Edge deviceNVIDIA Jetson TX2	Transfer learning	Python	Loss rate accuracy	High accuracy	Model complexity, accuracy degraded for largedataset

**Table 10 sensors-23-01639-t010:** Qualitative comparison of security and privacy in edge-based applications.

	Use Case	Ref	Contribution	AI Role (At the Edge)	AI Algorithm	Dataset	AI Placement	Employed Technology	Platform	Metrics	Benefits AI-Edge	Drawbacks
Security and privacy in edge-based applications	PP	[122]	Privacy-preserving-based secure C-ITS	Data encoding, prediction	LSTM-AE, RNN	ToN-IoT/CICIDS-2017	Fog nodes	Transfer learning	TensorFlow library, Keras	FAR-Accuracy-DR-PR, F1	Low communication overhead, low computation overhead, privacy preservation	Less scalability
		[123]	Privacy-preserving-based secure smart agriculture	Data encoding, Prediction	SAE, LSTM	ToN-IoT, IoT Botnet	Fog nodes	Transfer learning	TensorFlow library, Keras-	FAR-Accuracy-DR-PR, F1	Privacy preservation	Less scalability
		[124]	Improve the privacy of the user data	classification, adding noise	FCNN	ToN UCI knowledge discovery Archive database	Fog nodes	Transfer learning	Java Development toolkit (JDK) version 1.8, Weka	Scalability, processing time, accuracy	Higher scalability and efficiency	Fault tolerance
	AA	[8]	Enhance the security of MEC	Classification	DNN	Not mentioned	MEC	Transfer learning	Not mentioned	Computational cost, convergence speed	High convergence speed, low computational overhead	–
		[125]	Gait-based authentication to enhance security of mobile devices	Feature extraction-classification	CNN-LSTM	Matteo Gadaleta et al. dataset	Edge node/mobile	Transfer learning	Not mentioned	Complexity, accuracy	High accuracy	Energy consumption, memory not tested, limited dataset
Security and privacy in edge-based applications	ID	[126]	Distributed attack detection for IoT networks	Prediction	GRU-LSTM-CNN-DNN	NSL-KDD	Cloud, edge	Federated learning	TensorFlow	F1-score recall, detection time	Low response time, high accuracy, multiclass classification, scalability	Difficult retraining model at fog
		[127]	Real-time intrusion detection	Prediction	LSTM, GRU, CNN	CIDDS-01	Fog nodes	Transfer learning, SDN	Python	Accuracy, precision, recall, F1-score	Low accuracy, low response time, accuracy time, scalability	
		[128]	Real-time intrusion detection	AE, IF	NSL-KDD	Fog, cloud	Transfer learning	Python	Accuracy, precision, recall, F-measure value	High accuracy	–	
		[129]	Low-cost intrusion-detection system	Classification	SAE, mutual information (MI), C4.8 wrapper	Aegean WiFi Intrusion Dataset (AWID)	Edge device	Lightweight model	Not mentioned	FAR, accuracy, DR-PR, F1, MCC, TTB	High accuracy	Generalization not approved
		[130]	Real-time intrusion detection	Classification	DNN, PCA	BoT-IoT data set	Edge gateway (Raspberry Pi)-Cloud	Centralized, federated learning	Python	CPU usage, RAM usage, precision, F1-score, complexity	High accuracy, low complexity	Generalization not approved
		[131]	Real-time intrusion detection	Classification	Salp, LSTM	NSL-KDD, KYOTO, CICIDSCICIDS (AWS)	Fog gateway	Low-cost model	MATLAB	Accuracy	High accuracy, computational complexity	–
		[132]	Shamoon attack detection	Classification, feature extraction	K-means, KNN, PSO	Shamoon attack dataset	Fog nodes	Lightweight model	Not mentioned	Accuracy	Low computational cost	–
		[133]	Real-time attack detection	Classification, feature extraction	SVM, GWO	Opcode dataset	Edge server (PC)	/	TensorFlow	Computation time	High accuracy, high convergence	–

**Table 11 sensors-23-01639-t011:** The benefits of the integration of AI and edge in the eight reviewed domains.

Domain	Benefits of AI-Edge
Smart healthcare	Reduces latency and provides location-aware and real-time healthcare services.
Smart grid	Provides effective distribution and forecasting of energy
Smart agriculture	Provides powerful monitoring systems to help speed up the diagnosis and analysis of plants’ health conditions. Moreover, it helps to solve the problem of connectivity, monitor the statutes of the machine, and identify the fault in the machine in a timely manner
Smart environment	Improves data quality, reduces computational modeling complexity, and improves the mining efficiency of ocean big data. For air-quality monitoring, considering regional characteristics when distributing various site-monitoring models enhances the performance of monitoring
Security and privacy	Increases security and privacy by adding noise and encryption to data, and distinguishing legitimate edge nodes from malicious nodes and attackers
Smart industry	Provides immediate services to customers with minimal delays and errors; it also helps in detecting the credit risks of legitimate customers and detecting and preventing fraudulent activity
Smart transportation	Manages real-time parking, traffic flow prediction, and supports intelligent mobility decisions
Smart education	Improves online and real-time course management services, addresses poor portability of the experience, and improves distance learning

## Data Availability

Not applicable.

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
