# Peer review of "At the Confluence of Artificial Intelligence and Edge Computing in IoT-Based Applications: A Review and New Perspectives"

_sensors, 2023, doi:10.3390/s23031639_

Round 1

Reviewer 1 Report

This paper presents a comprehensive review on artificial intelligence and edge computing in IoT-based applications. Overall this manuscript is well-written. Some comments are as follows.

1. The authors may want to summarize the existing challenges and unsolved problems in IoT-based applications after the first paragraph. Then introduce the importance of AI and edge computing for solving these challenges.

2. The two paragraphs in lines 60-67 in Page 2 can be combined into one paragraph.

3. "We review related works that have been published from 2019 to the present" in Line 90, Page 3. A specific number of the studies reviewed in this paper can be clarified.

4. The paragraph in Lines 537-541 in Page 17 is not finished.

5. The reference number in Line 1109 Page 36 "using cryptographic algorithms" is missing.

6. For "Spatio-temporal correlated data issue" in Line 1154 Page 37, the authors should point out that graph-based deep learning is a promising solution with the following references:

* Li Y, Xie S, Wan Z, et al. Graph-powered learning methods in the Internet of Things: A survey. Machine Learning with Applications, 2023, 11: 100441.

* Jiang W. Graph-based Deep Learning for Communication Networks: A Survey. Computer Communications, 2022, 185:40-54.

7. The discussion for Section 4.3 Scalability is not comprehensive and should be enhanced with specific methods.

Author Response

please, see the attached file

Reviewer 2 Report

At the confluence of Artificial Intelligence and Edge computing in IoT-based Applications: A review and new perspectives

1. At the start, the type is article is mentioned, this should be review type.

2. In line 68, the authors have written that "Several review papers have investigated the integration of AI in edge-based applications" How your contributions are unique as several potential publications are already available? 

3. Some comparative tables can be seen in the manuscript; the tables should use more columns in terms of more parameters to illustrate the facts in more detail. 

4. Only two figures are used in the entire manuscript to highlight the key facts some more detailed images and one particular detail flow chart should be used to highlight the entire manuscript's findings.

5. Section 3:-Discussions of related works: findings and insights: requires major revisions since this is the backbone of the entire manuscript so the tabular representation of the findings should be preferred, some graphs should be drawn, only theoretical discussions severely weakened the quality of the manuscript though one table has been presented but it does not cover the purposes fully.

6. Second important pillar of the manuscript is the Section4: Open issues and future research directions; Potential technical discussions should be discussed here, again latest trends with more refined way should be represented with brief examples etc., some diagrams will surely enhance the quality of the manuscript, Section 4.2 some issues are highlighted such as

1. Limited computational power for DL training

2. Deep learning overfitting at the edge

3. Distributed learning/ Federated learning

There are generalized issues and every potential researcher in this area is aware of these issues, so some more refined issues need to be discussed with proper examples.

7. Finally, the conclusion section also requires refinement with the precise illustration of key points only.

Overall, a lot of reformations are required to enhance the quality of the manuscript.

Author Response

please, see the attached file

Reviewer 3 Report

Please find the attachment

Author Response

please, see the attached file

Round 2

Reviewer 1 Report

Dear authors,

Thanks for revising and resubmitting the manuscript. It has improved a lot and no further comments.

Author Response

Please, see the file attached

Reviewer 2 Report

At the confluence of Artificial Intelligence and Edge computing in IoT-based Applications: A review and new perspectives

Two specific points should be considered

1. Figure 10 should be re-designed.

2. Add a few more references from 2022.

Author Response

Please, see the file attached

Reviewer 3 Report

Authors have made all the changes as suggested. Hence paper is accepted in its present form.

Author Response

Please, see the file attached
